

# Improving snowfall representation in climate simulations via statistical models informed by air temperature and total precipitation

Flavio Maria Emanuele Pons[1] and Davide Faranda[1,2]

[1]LSCE-IPSL, CEA Saclay l'Orme des Merisiers, CNRS UMR 8212 CEA-CNRS-UVSQ, Université Paris-Saclay, 91191 Gif-sur-Yvette, France
[2]London Mathematical Laboratory, 14 Buckingham Street, London, WC2N 6DF, UK
*flavio.pons@lsce.ipsl.fr*

**Correspondence:** Flavio Pons (flavio.pons@lsce.ipsl.fr)

**Abstract.** The description and analysis of compound extremes affecting mid and high latitudes in the winter requires an accurate estimation of snowfall. Such variable is often missing for in-situ observations, and biased in climate model outputs, both in magnitude and number of events. While climate models can be adjusted using bias correction (BC), snowfall presents additional challenges compared to other variables, preventing from applying traditional univariate BC methods. We extend
the existing literature on the estimation of the snowfall fraction from near-surface temperature, which usually involves binary thresholds or fitting parametric nonlinear functions. We show that, combining breakpoint search algorithms to define threshold temperatures and segmented regression models, it is possible to obtain accurate out-of-sample estimates of snowfall over Europe in ERA5 reanalysis, and to perform effective BC on the IPSL-WRF high resolution EURO-CORDEX climate model only relying on bias adjusted temperature and precipitation. This method offers a feasible way to reconstruct or adjust snowfall
observations without requiring multivariate or conditional bias correction and stochastic generation of unobserved events.

## 1  Introduction

Despite the expectations of less frequent snow events in a warming climate, there are still several motivations to study trends in future snowfall. First of all, snowfall extremes can still have a great impact on economy and society. Recent snowfalls over large populated areas of France in February 2018 or in Italy in January 2017 caused transport disruption, several casualties
and economical damages. Snow is also an important hydrological quantity and a touristic resource. Although climate models predict a general reduction in snowfall amounts due to global warming, accurate estimates of this decline heavily depend on the considered model. Large discrepancies in snowfall amounts indeed exist for observational or reanalysis datasets: in detecting recent trends in extreme snowfall events, Faranda (2020) has also investigated the agreement between the ERA5 reanalysis and the E-OBSv20.0e gridded observations in representing snowfall, considering snowfall all precipitation that occurred on days
where the average temperature was below 2 °C. He found that observed trends and the agreement in absolute value between the two datasets largely depended on the considered region. Overall, the climatologies of snowfall provided by the two datasets had similar ranges, although ERA5 tended to overestimate snowfall compared to E-OBSv20.0e. Even though such binary





separation of snowfall using a temperature threshold seemed a good option to retrieve snowfall data from E-OBSv20.0e, it has obvious limitations, especially in reproducing the entire probability distribution rather than only long run snow totals. In this

manuscript we explore the possibility of reconstructing snowfall from temperature and precipitation via adequate statistical models, instead of using the snowfall provided by climate models, which is subject to heavy parametrizations.

Climate Models are the primary tool to simulate multi-decadal climate dynamics and to generate and understand global climate change projections under different future emission scenarios. Both regional and global climate models have coarse resolution and contain several physical and mathematical simplifications that make the simulation of the climate system com-

putationally feasible, but also introduce a certain level of approximation. This results in statistical biases that can be easily observed when comparing the simulated climate to observations or reanalysis datasets. Therefore, they provide limited actionable information at the regional and local spatial scales. To circumvent this problem, it is of crucial importance to correct these biases for impact and adaptation studies and for the assessment of meteorological extreme events in a climate perspective, (see e.g. Ayar et al., 2016; Grouillet et al., 2016).

Bias correction attempts to resolve the scale discrepancy between climate change scenarios and biases detected by comparing simulations to observation required for impact assessment. Two main downscaling approaches have been developed since the early 1990s: Dynamical downscaling (based on Regional Climate Models, RCMs) and Statistical Downscaling Models (SDMs), which are nowadays recognized as complementary in many practical applications. In practice, the BC step usually consists of a methodology designed to adjust specific statistical properties of the simulated climate variables towards the

observed climatology. The chosen statistics can be very simple, e.g. mean and variance, or include dynamical features in time, such as a certain number of lags of the autocorrelation function for time series data; they can focus on a limited number of moments or aim at correcting the entire probability distribution of the observable; the correction can also be carried out in the frequency domain, so that the entire time dependence structure is preserved. For an overview of various BC methodologies applied to climate models see, for example, Teutschbein and Seibert (2012, 2013); Maraun (2016). Key efforts have been

also put on specific BC developments for precipitation (Vrac et al., 2016) and for multivariate (i.e., multi-sites and variables) approaches, both in downscaling and BC contexts (Vrac and Friederichs, 2015; Vrac, 2018).

Despite the effort devoted to correcting precipitation bias, only few studies propose specific BC methods for snowfall data from climate projections. For instance, Frei et al. (2018) propose a bias adjustment of snowfall in EURO-CORDEX models specific to the Alpine region, involving altitude-adjusted total precipitation and a single threshold temperature to separate rain

and snow. Krasting et al. (2013) study snowfall in CMIP5 models at the Northern emispheric level, highlighting biases but without suggesting any methodology to reduce them, while Lange (2019) proposes a quantile mapping approach that can be used for univariate BC of snowfall. On the other hand, most efforts are focused on correcting observationa (Karbalaee et al., 2017; Wang et al., 2017; Naseer et al., 2019; Panahi and Behrangi, 2019) or reanalysis data (Cucchi et al., 2020; Panahi and Behrangi, 2019).

Indeed, snowfall presents additional challenges compared to other variables, preventing from obtaining accurate results by means of traditional univariate BC methods. Besides the discontinuity of snowfall fields - a feature in common with total precipitation - snowfall is the result of complex processes which involve not only the formation of precipitation, but also the


existence and persistence of thermodynamic conditions that allow for the precipitation reaching the ground in the solid state. As
a result, snow is often observed in mixed phase with rain, especially when considering daily data. This phase transition poses

additional challenges to the bias correction of snowfall, namely the need of separating the snow fraction, using the information
provided with temperature data. All these issues would require the application of multivariate/conditional BC methodologies,
introducing heavy complications and non-trivial issues (François et al., 2020).

In the following, $T$ denotes the mean daily near-surface temperature, $P_{tot}$ the total daily precipitation, $SF$ the total daily
snowfall, $f_s$ denotes the snow fraction and $f_r$ the rain fraction of total precipitation.

The fact that $T$ is an effective predictor of $f_s$ was first observed by Murray (1952). This study tried to link the precipitation
phase also to other variables, such as the freezing level and the thickness of pressure difference layers (1000-700 hPa and 1000-
500 hPa), finding that near-surface temperature alone is as effective in predicting the snow fraction as the others. Following
this result, several authors suggest the use of a binary separation of the snow fraction based on a threshold temperature both
in climatological (US Army Corps of Engineers, 1956; de Vries et al., 2014; Zubler et al., 2014; Schmucki et al., 2015) and

hydrological (Bergström and Singh, 1995; Kite, 1995) studies. In this setting, a threshold temperature $T^*$ is chosen, so that the
fraction $f_s$ of total precipitation falling as snow is:

$$\text{for } T \leq T^* : \qquad f_s = 1$$
$$\text{for } T > T^* : \qquad f_s = 0$$

Pipes and Quick (1977), in a hydrological modelling context, propose a double threshold linear interpolation:

$$\text{for } T \leq T^*_{low} : \qquad f_s = 1$$
$$\text{for } T^*_{low} < T < T^*_{high} : \qquad f_s = 1 - \frac{T - T^*_{low}}{T^*_{high} - T^*_{low}}$$
$$\text{for } T > T^*_{high} : \qquad f_s = 0$$

This choice constitutes a simple way to give a realistic representation of the relationship between the snow fraction and near-
surface temperature, which often resembles an inverse S-shaped curve. L'hôte et al. (2005) find similar relationships over the

Alps and the Andes, pointing to a broad validity of such an assumption; several other instances of research finding evidence of
such relationship are also mentioned in the following paragraph. An important limitation of this method is that the thresholds
$T_{low}$ and $T_{high}$ are fixed; As reported by Kienzle (2008), Pipes and Quick (1977) use threshold values $T^*_{low} = 0.6^{\circ}$C and
$T^*_{high} = 3.6^{\circ}$C to estimate snowfall in the US. Wen et al. (2013) points out that a similar method had already been implemented
by US Army Corps of Engineers (1956), using three different threshold values; this adds parameters to the model, which then

requires even finer tuning before being applied to prediction. In more recent years, the double threshold method has also been
applied to climatological analysis, for example in McCabe and Wolock (2008, 2009), where the authors stress that the choice
of the thresholds require an important calibration procedure step. Still in the class of threshold models, Kienzle (2008) uses a
parameterization considering a temperature $T_T$ at which the precipitation falls half as rain and half as snow, and a temperature
range $T_R$ within which both phases co-exist. This method requires validation using reliable and sub-daily station data, making

it less suited for the characterization of gridded snowfall over large domains in reanalysis or climate models.





Slightly more complex methods aim at reproducing the quasi-smooth shape of the precipitation phase transition by fitting S-shaped functions to the relationship between $T$ and $f_s$ (or $f_r$). For example, Dai (2008) proposes a hyperbolic tangent $f_s = a[\tanh(b(T-c))-d]$, while McAfee et al. (2014) choose a logistic function $f_s = (1+e^{-a+bT})^{-1}$, both fitted via nonlinear least squares (NLS). Harder and Pomeroy (2013) propose a similar procedure, adopted also by Pan et al. (2016), based on a

sigmoidal function $f_r = (1+b \cdot c^{T_i})^{-1}$, where $b$ and $c$ are parameters calibrated using data from a single location, and $T_i$ is the so-called hydrometeor temperature, i.e. the temperature at the surface of a falling hydrometeor, defined by Harder and Pomeroy (2013) as a function of air temperature and humidity. While Harder and Pomeroy (2013) find that this method provides more accurate results compared to simple and double thresholds, the estimation of $T_i$ requires reliable measurements or predictions of relative humidity, making this technique more suited to treat observational data.

Wen et al. (2013) present a comparison of some of these methods. In particular, they test the capability to reproduce the snow fraction in a regional model with no atmosphere-surface coupling forced by observation, and in a coupled regional circulation model with land interaction. The methods put to the test are: single threshold (with $T^*$ taken to be $0^{\circ}$C and $2.5^{\circ}$C); double threshold, both with the parameter values fixed by US Army Corps of Engineers (1956) and Pipes and Quick (1977); the model proposed by Kienzle (2008) and the nonlinear relationship specified by Dai (2008). Results are mixed, with different methods

performing differently in the two models. It is worth to stress that Wen et al. (2013) do not tune the parameterizations, nor they assess whether the chosen single threshold is optimal for the considered datasets.

We aim at finding a feasible method that allows for accurate estimation of $f_s$ as a function of $T$, and then of $SF = f_s \cdot P_{tot}$ in gridded time series datasets, overcoming the drawbacks of the methodologies applied so far in the literature. First, by proposing a method to detect candidate values for the threshold temperature(s) in an automated and computationally feasible way; second,

by fitting nonlinear functions that can incorporate the threshold value(s) without assuming parametric specifications (such as hyperbolic tangent or logistic function) that may not be flexible enough to describe the phenomenon at sensibly different locations. This way, we aim at ensuring sufficient flexibility to apply model results to out-of-sample data, while adopting very simple model specifications informed by the retrieved threshold values.

The rest of the paper is organised as follows: in Section 2 we describe the datasets used for model specification and as-

sessment of the snowfall reconstruction performance; in Section 3 we illustrate the methodologies used to define threshold temperatures, model the snow fraction and select the best model. In Section **??** we present the results obtained on the considered datasets, including case studies for two highly sensitive regions; finally, in Section 5 we discuss our conclusions.

## 2    Data

### 2.1    The ERA5 Reanalysis Dataset

Most of the hydrological and climatological studies cited in Section 1 are focused on limited areas where snowfall is a recurrent phenomenon. In general, it is possible to find high quality snowfall data for areas heavily affected by frequent snowfall, such as Scandinavian countries and the Alpine region (Auer et al., 2005; Scherrer and Appenzeller, 2006; Isotta et al., 2014) which,


however, can still suffer from the lack of reliable data at high altitudes (Beaumet et al., 2020). On the other hand, good quality snow data at the synoptic scale are in general difficult to obtain (Rasmussen et al., 2012).

Our goal is to elaborate a methodology that allows reconstructing snowfall at large scales and extending the subsequent climatological analysis beyond mountain areas or high latitudes, including also regions where these phenomena are relatively rare, and occasional extremes can cause service disruption, damage, or economic and human loss. Therefore, we decide to rely on a gridded reanalysis dataset at the European scale to specify and validate our snowfall model, rather than on observational data from limited areas.

In particular, we use the Reanalysis 5th Generation product (ERA5) provided by the European Centre for Medium-Range Weather Forecast (ECMWF). This dataset has a high (0.25°) horizontal resolution over Europe and accurate physical parameterizations (Copernicus Climate Change Service, 2017) over the period 1979-2018. We consider daily data over a domain covering the area between 26° and 70° North and between -22° to 46° East, consisting of a lat-lon grid of $273 \times 177$ points covering Europe, part of the Eastern Atlantic and parts of Russia and North Africa. We only include in our analysis December,

January and February (DJF) days.

    In most reanalysis and climate simulation models, snowfall is represented as snowfall flux $SF$ in $\mathrm{kg\,m^{-2}\,s^{-1}}$ (Copernicus Climate Change Service, 2017, see also `https://esgf-node.ipsl.upmc.fr` for CMIP5, CMIP6 and EURO-CORDEX variable lists), from which it is easy to retrieve total snowfall through time integration. Here we consider $SF$ expressed in $\mathrm{m\,day^{-1}}$ of equivalent water depth, considering that $1\,\mathrm{kg\,m^{-2}}$ corresponds to $10^{-3}$ m of water equivalent. This

quantity is relevant for hydrologists, being closely related to runoff and river discharge, but also for climatologists, since it well represents the intensity of the phenomenon. However, it is worth to mention that, as a main drawback, $SF$ does not distinguish whether a precipitation event produces or not accumulation on the ground.

    Snowfall in ERA5 consists of snow produced by both large-scale atmospheric flow and convective precipitations. It measures the total amount of water accumulated during the considered time step as the depth of the water resulting if all the snow melted

and was spread evenly over the grid box. We start from hourly snowfall flux and construct the daily $SF$ by summing up the snowfall over intervals of 24 hours. We chose ERA5 dataset as the preferential one for our study because of its physical consistency and the use of advanced assimilation techniques for its compilation (Faranda (2020)).

### 2.2   Historical Climate Simulation

In this paper, we use outputs from the IPSL_WRF climate projection model from the EURO-CORDEX project (Vautard et al.,

2020). In particular, we consider DJF data from climate simulations of the historical period 1979-2005 over the same domain and grid step as the reanalysis dataset described in Section 2.1. The relevant variables are $T$, $P_{tot}$ and $SF$, sampled at daily time step. The datasets are freely available via the Earth System Grid Federation (ESGF) nodes (https://esgf.llnl.gov/nodes.html). Both $T$ and $P_{tot}$ are available in a bias adjusted version as described by Bartók et al. (2019), using the cumulative distribution function transform (CDF-t) introduced by Vrac et al. (2012) and further developed by (Vrac et al., 2016) to improve

the adjustment of precipitation frequency. CDF-t is a distribution mapping method, frequently chosen in studies that involve





climate projections, as they perform better than methods based on linear transformations in case of changing future conditions (Teutschbein and Seibert, 2013).

In the following, we show how statistical modelling of $SF$ based on bias adjusted $P_{tot}$ and $T$ can replace direct BC of the snowfall, markedly improving model $SF$ statistics with respect to the reference data. For both ERA5 and IPSL_WRF we apply
a binary land-sea mask to only consider snowfall over the continents.

## 3    Methods

### 3.1    Statistical modelling of the snow fraction

In this Section, we describe a set of candidate models for the snow fraction $f_s$ as a function of the near surface temperature $T$ and how we compare their performance in terms of accurate reconstruction of the snowfall in an out-of-sample test set. Even
though the statistical models consider the snow fraction $f_s$ as the target variable, models are mainly validated and compared based on the capability to reproduce total snowfall, $SF = f_s \cdot P_{tot}$.

First, we consider the single threshold method (STM) as our naive baseline model. Given the spatial extent of our dataset and the relatively fine grid resolution, we believe that more refined models could be better suited to the purpose of climatological analysis of snowfall. In particular, we aim at finding parsimonious models that can be easily fitted point-wise on the grid,
producing location-specific parameter estimates that we may exploit to extract information about the phenomenon.

We decide not to follow methods based on directly fitting S-shaped functions to the data, such as in Dai (2008); Harder and Pomeroy (2013); McAfee et al. (2014). While these models can be fitted point-wise, the parameter estimation must be carried out using NLS, a method that can be very sensitive to assigned initial parameter values, numerical algorithm, and convergence criterion. Moreover, as it will be detailed in Section 4, the assumption of an inverse S-shaped relationship between $T$ and $f_s$
is not realistic for all areas in the EURO-CORDEX domain. This implies that forcing the fit of a fixed S-shaped function may result in a poor fitting and forecasting performance.

Instead, we rely on a more flexible framework, consisting of two steps. First, for each grid point we analyse the relationship between $T$ and $f_s$ and we exploit a breakpoint (or changepoint) search algorithm to assess whether two, one or no thresholds should be assumed to describe the rain-snow transition as a function of near-surface temperature. Then, we rely on regression
to fit grid point-specific statistical models of the snow fraction, incorporating the information about threshold temperatures.

In all the regression models discussed in the following, but not in the STM, we use the standardized temperature anomalies as independent variable. It is important to remark that this requires computing standardized anomalies using the same mean and standard deviation as in the reference data. This is not an issue when considering BC temperature in the historical period of a climate model, since CDF-t BC guarantees a good matching of the first two moments of the probability distribution of $T$.
However, it requires attention when considering climate projections under future emission scenarios, where the standardized anomalies should still be computed with respect to the reference historical period, to avoid losing climate change signal.



### 3.1.1 Single Threshold Model (STM)

First of all, we assess the results obtained applying the STM, introduced by Murray (1952). As already discussed in Section 1, this method has been used in both hydrological and climatological contexts for almost 70 years, until present. The most
difficult step of this specification, and the greatest drawback of the method, is the choice of the threshold temperature itself.

Despite its simplicity, this technique presents some advantages. First, if $T^*$ is accurately chosen to preserve long run snow totals in the reference dataset, also snow totals in the climate model are not expected to be severely biased. Moreover, this method can potentially represent extreme DJF snowfall in cold climates better than more complex models, since it is very likely that important snowfall episodes happen below the threshold daily temperature and correspond to events during which
the totality of daily precipitation falls as snow. On the other hand, heavy wet snowfall with disruptive effects is a well known phenomenon in temperate climates (Nikolov and Wichura, 2009; Bonelli et al., 2011; Llasat et al., 2014; Ducloux and Nygaard, 2014).

However, this method is also naive, as it gives a binary representation of a quantity continuously varying in [0,1]: this makes it impossible for the model to provide insights on snowfall features in case of more in-depth climatological analysis or more
refined hydrological models. Furthermore, the search for the optimal threshold should not be complex or computationally expensive, otherwise invalidating the advantage of using such simplified assumptions. This does not prevent from detecting a representative value of $T^*$ when conducing a site-specific or local scale study, but a single value of the threshold extracted from the literature or estimated considering the whole aggregated dataset can be a gross approximation in case of gridded high resolution data on a wide domain such as the EURO-CORDEX one.

Estimates reported in the literature for the single threshold range across quite different values. For example, Auer Jr (1974) finds the optimal temperature to give a binary representation of the snowfall to be $2.5°C$ analysing station data in the U.S., but values as low as $0°C$ are reported by Wen et al. (2013). In an analysis of snowfall trends over Europe in the last decades, based on ERA5 and E-OBS data, Faranda (2020) suggests a threshold $T^* = 2°C$, also finding that any threshold between 0 and $2.5°C$ does not significantly change overall results in that specific study, focused on observed trends in snowfalls during
the last decades.

However, it is worth to mention that the results by Faranda (2020) refer to snowfall over Europe where spatial averaging is applied at regional or country level: it is possible that different thresholds in the interval 0 - $2.5°C$ are more suited for different parts of the domain, but the errors cancel out in the spatial averaging thus not showing sensitivity to the threshold value in terms of long run statistics. This condition does not hold if the reconstruction of the snowfall must be carried out
preserving spatial structures, and in general we do not expect to be able to obtain an accurate representation of the snow fraction using a single value for the threshold over the whole domain. On the other hand, we can still expect this method to perform conveniently when considering long term spatially averaged statistics, but also right-tail extremes in cold climates or elevated locations: here, extreme snowfall leading to important snow accumulation on the ground is expected to be concurrent with large daily precipitation and low temperature, and then with high values of $f_s$. Choosing a threshold above the freezing





point and considering any precipitation happening with any daily mean temperature lower than such thresholds is conservative in terms of estimation of extreme events, despite its inadequacy in terms of reproducing more complex features of snowfall.

In the following, we discuss a methodology that encompasses the case of a locally selected threshold temperature, while enabling to determine the optimal number of thresholds and their respective values for each point of the considered domain.

### 3.1.2 Multiple Threshold Regression (MTR)

In order to overcome the limitations of the STM of $f_s$, we aim at reproducing the potentially nonlinear relationship between $T$ and $f_s$. As already mentioned in Section 1, we decide not to adopt NLS to directly fit S-shaped functions to the data. Other than the sensitivity of this methodology to the optimization algorithm and to initial values, we envisage two more reasons to avoid direct S-shape fitting. First, there is no prior indication of which among the possible S-shaped functions (logistic, hyperbolic tangent, sigmoid) is universally better; this would require to compare the fit and the predictive performance of

different specifications at each grid point. Moreover, the points separating the asymptotically horizontal regimes from the centre of the S-shaped curve can be seen as corresponding to two temperature thresholds analogous to $T_{low}^*$ and $T_{high}^*$ in Pipes and Quick (1977). While the values of these threshold temperatures carry interesting information, it is not immediate to retrieve them from the specified S-shaped functions, nor it is to inform the NLS estimates with these threshold values if their estimate is available.

Essentially, we propose a way to extend the method by Pipes and Quick (1977). First, we determine the optimal number $m$ of thresholds temperatures for each grid point and their value. Then, in each of the $m + 1$ regimes corresponding to the estimated $m$ thresholds, we describe the relationship between $T$ and $f_s$ using a regression model. In the following, we describe the search algorithm used to determine the threshold temperatures at each grid point, and we show three different ways to perform MTR: (i) segmented regression on logit-transformed data, (ii) beta regression with logit link function, and (iii) spline regression on

logit-transformed data.

**Breakpoint analysis and segmented regression**

In order to estimate the temperature thresholds, we rely on breakpoint analysis. This method was originally developed by Bai (1994) to detect and date a structural change in a linear time series model and later extended to the case of a time series with multiple structural breaks (Bai and Perron, 1998). The technique was further generalised by Bai and Perron (2003) to

the simultaneous estimation of multiple breakpoints in multivariate regression. In the following, we rely on this formulation, implemented by Zeileis et al. (2003) in the R package `strucchange` (Kleiber et al. (2002)).

The method can be summarised as follows. Let us first consider the case of a univariate response variable $y_i$ and a $k \times 1$ vector of explanatory variables $x_i$, evolving over time $i = 1, \ldots, n$ and linked by a linear relationship:

$$y_i \approx x_i^T \beta_i + \varepsilon_i \qquad (i = 1, \ldots, n), \tag{1}$$

where $\beta_i$ is a $k \times 1$ vector of regression coefficients and $\varepsilon_i$ a random term. The first component of $x$ is the unit, so that the first component of $\beta$ is the model intercept. The null hypothesis when testing for structural change is that the slope $\beta_i$ is constant




over time, i.e.

$$H_0: \quad \beta_i = \beta_0 \quad \forall i = 1, \ldots, n \tag{2}$$

against the alternative hypothesis that there is at least one date $i$ such that $\beta_i \neq \beta_0$. In case there are $m$ breakpoints, there are

$m+1$ regimes, i.e. time segments over which the regression coefficient is constant, and the model can be written as

$$y_i \approx x_i^T \beta_j + \varepsilon_i \quad (j = 1, \ldots, m+1; \ i = i_{j-1}+1, \ldots, i_j), \tag{3}$$

where $j$ denotes the segments and $\mathcal{I}_{m,n} = \{i_1, \ldots, i_m\}$ is the m-partition representing the set of breakpoints, with $i_0 = 0$ and $i_{m+1} = n$ by convention. The null hypothesis in Eq. (2) is tested in a generalized fluctuation framework: model in Eq. (1) is fitted, and the model residuals $\hat{\varepsilon}_i$ are used to construct an empirical process that captures their fluctuations, provided that a

functional central limit theorem holds. In general, this requires that $x_i$ is stationary and $\varepsilon_i$ is a martingale difference sequence independent on $x_i$. However, the method implemented by Zeileis et al. (2003) allows for less stringent conditions, and in particular the stationarity of $x_i$ can be relaxed to admit trending independent variables.

It is crucial to remark that, while the method was originally developed to detect structural breaks in time series, $i$ must not necessarily represent time. In our case, $x$ is near-surface temperature and $y$ the snow fraction: the breakpoints in the scatterplot

of $y$ against $x$ can indeed be interpreted as threshold temperature values. Suppose that an m-partition is given and model 3 is estimated using ordinary least square (OLS) regression, giving a total residual sum of squares:

$$RSS(i_1, \ldots, i_m) = \sum_{j=1}^{m+1} \sum_{l=i_{j-1}+1}^{i_j} \hat{\varepsilon}_k^2 = \sum_{j=1}^{m+1} rss(i_{j-1}+1, i_j), \tag{4}$$

where $rss(i_{j-1}+1, i_j)$ is the residual sum of squares in the $j$th segment of the partition. Then, finding the breakpoints in model 3 consists of finding the set of points $\hat{i}_1, \ldots, \hat{i}_m$ such that

$$(\hat{i}_1, \ldots, \hat{i}_m) = \underset{i_1, \ldots, i_m}{\arg\min} \, RSS(i_1, \ldots, i_m). \tag{5}$$

This operation is performed using the linear programming method illustrated in Bai and Perron (2003), which is of order $O(n^2)$ for a sample size $n$ and any number of breakpoints $m$. The optimal number of breakpoints can be estimated as well, by considering several possible values of $m$ and choosing the one that results in the smallest $RSS(i_1, \ldots, i_m)$. Since we aim at reproducing an S-shaped relationship between $T$ and $f_s$, which finds strong support in the literature, we assume the existence

of at most two threshold temperatures, dividing three regimes in analogy to Pipes and Quick (1977). However, for higher flexibility, we will not necessarily assume a complete saturation of snowfall and rainfall below and above the two thresholds, allowing $\beta_i \neq 0$ in all regimes.

In principle, imposing two threshold temperatures is not necessarily the best assumption for every point on the grid. In fact, the EURO-CORDEX domain clearly includes areas where, even only considering DJF precipitation, snowfall is infrequent and

usually happens at positive temperatures, as well as areas (such as Scandinavia and continental Eastern Europe) where most of winter precipitation is likely to fall as snow. For this reason, we admit $m = 0, 1, 2$ as possible numbers of thresholds at each





point, and we use the breakpoint search described above to determine both $m$ and the corresponding threshold temperatures, if any.

Once the number of thresholds and their values are obtained, we estimate a segmented linear regression of the general form presented in Eq. (3). First, it is worth to mention the assumptions required for a consistent parameter estimation in the context of a simple linear model as in Eq. (1). For the sake of simplicity, let us assume a model without intercept, so that $x_i$ reduces to a one-dimensional random variable. The expected value of the response variable is

$$\mu_i = E[y_i] = x_i\beta$$

so that $y_i \approx \mu_i + \varepsilon_i$. If $\varepsilon_i$ is a sequence of mutually independent zero mean and homoskedastic (i.e. with constant variance $\sigma^2$) random variables, fitted values can be written as $\hat{\mu}_i = \hat{\beta}x_i$, where the estimator $\hat{\beta}$ is obtained via OLS:

$$hat\beta = \frac{\sum x_i y_i}{\sum x_i^2}.$$

It is easy to prove (see, for example, Wood (2017)) that $E[\hat{\beta}] = \beta$ and $Var[\hat{\beta}] = \sigma^2/\sum x_i^2$. If $\varepsilon_i$ are not only homoskedastic and independent, but also normally distributed, $\varepsilon_i \sim N(0, \sigma^2)$, the estimator $\hat{\beta}$ is normally distributed around the true value,

$$\hat{\beta} \sim N\left(\beta, \frac{\sigma^2}{\sum x_i^2}\right)$$

so that it is possible to make inference on $\hat{\beta}$, for example to test significance and construct confidence intervals. Assuming that the random term $\varepsilon_i$ has a normal distribution also implies that $y_i \sim N(x_i\beta, \sigma^2)$. Under these assumptions, OLS estimates of $\beta$ coincide with maximum likelihood estimates. A check of how realistic these assumptions are can be done by considering the model residuals $\hat{\varepsilon}_i = y_i - x_i\hat{\beta}$ and testing for normality, autocorrelation and homoskedasticity. For the sake of greater readability, in the following we omit the time index $i$ unless necessary.

In our case, the dependent variable is the snow fraction $f_s$ and the explanatory variable $x$ is the near-surface temperature $T$. Since $0 \leq f_s \leq 1$, the assumptions required to estimate the regression coefficient using OLS are not met, even approximately. The problem can be regularized using the logit function of the snowfall fraction, so that the transformed variable can assume any real value:

$$f_s' = \text{logit}(f_s) = \log\left(\frac{f_s}{1 - f_s}\right). \tag{6}$$

The logit function is a natural transformation for binary variables or variables assuming values in [0,1], and is then used as the canonical link in case of generalised linear models with binary response (Agresti, 2015) and for beta regression (Smithson and Verkuilen, 2006), as also discussed in Section 3.1.3. Notice that the quantity $f_s/(1-f_s)$ represents the odds of snowfall against rainfall, and it is a positive quantity without upper limit, characterized by positive skewness, and it is sometimes assumed that its logarithm approximately follows a normal distribution (Bland and Altman, 2000).

Given $m$ thresholds, $m+1$ regression models must be estimated. If no threshold is found, the problem reduces to a standard linear regression model:

$$f_s' \approx \beta_1^0 + \beta_1 T + \varepsilon_1, \tag{7}$$





where $\beta^0$ denotes the intercept and the index $i$ was omitted for simplicity. If only one threshold temperature $T^*$ is detected we write,

for $\quad T \leq T^*: \qquad f'_s \approx \beta_1^0 + \beta_1 T + \varepsilon_1$


for $\quad T > T^*: \qquad f'_s \approx \beta_2^0 + \beta_2 T + \varepsilon_2$

(8)

and for grid points where two thresholds are found:

for $\qquad T \leq T^*_{low}: \qquad f'_s \approx \beta_1^0 + \beta_1 T + \varepsilon_1$

for $\quad T^*_{low} < T < T^*_{high}: \qquad f'_s \approx \beta_2^0 + \beta_2 T + \varepsilon_2$

(9)

for $\qquad T > T^*_{high}: \qquad f'_s \approx \beta_3^0 + \beta_3 T + \varepsilon_3$

In the following, the symbol $\beta$ without any further indexing will denote the complete parameter vector, i.e. $\beta = (\beta_1^0, \beta_3^0, \beta_3^0, \beta_1, \beta_3, \beta_3)$ or any of the subsets defined by the model assumed among the ones described by Eq.s (7)-(9), and $\hat{\beta}$ will denote the vector of

estimates.

A model of the type 8 or 9 is known as *segmented regression* and the slope in each regime can be estimated with OLS if the assumptions for simple linear regression are (at least approximately) met. In particular, we refer to the methodology described in Muggeo (2003). Let $\mu = E[y]$ denote the expected value of the response variable, $g(\cdot)$ a link function and $\eta(x)$ the linear predictor: in a generalized linear model framework, $g(E[y]) = \eta(x)$, and in the case of simple linear regression $g(\cdot)$ is the

identity function. In case there is one threshold $T^*$, the relationship between the mean response $\mu$ and the explanatory variables can be modelled by adding nonlinear extra terms to the predictor:

$$\mu_i = \eta(x_i) + \beta_1 x_i + \beta_d (x_i - T^*) I(x_i > T^*), \tag{10}$$

where $I(\cdot)$ denotes the indicator function. Here, $\beta_1$ is the slope in the first regime, while $\beta_d$ is the difference-in-slopes between the two regimes separated by $T^*$. Muggeo (2003) shows that the nonlinear term in Eq. (10) can be approximated by a linear

representation, enabling the use of standard linear regression to estimate the parameters. In particular, provided a first estimate of the threshold $T^*$, Eq. (10) is estimated by fitting iteratively a linear model with a predictor including an additional term:

$$\mu = \eta(T_i) + \beta_1 x_i + \beta_d (T_i - T^*) I(T_i > T^*) - \gamma I(T_i < T^*), \tag{11}$$

where $\gamma$ measures the gap between the two regression lines estimated in the regimes located at the left and right of $T^*$. At each iteration, the estimated values of the gap, $\hat{\gamma}$, and of the difference-in-slope, $\hat{\beta}_d$, are used to update the previous value of the

threshold, so that the new value is $\tilde{T}^* = T^* + \hat{\gamma}/\hat{\beta}_d$. When $\hat{\gamma} \approx 0$ the algorithm converges and the values $\tilde{T}^*$ and $\hat{\beta}_d$ (and then $\hat{\beta}_2$) are saved. The method is easily generalized to the case $m$ breakpoints, and is implemented in the R package `segmented` (Muggeo et al., 2008). In the following, $T^*$ will denote the threshold temperatures estimated using the breakpoint analysis (Eq. (5)), and $\tilde{T}^*$ the estimates updated in the segmented regression procedure (Eq. (11)). In practical terms, the procedure is conducted as follows for each grid point.

(i) The number and values of threshold temperatures are found using the `breakpoints()` function from the R package `strucchange`;





(ii) a threshold-free model of the form 7 is estimated via OLS using the `lm()` function from the native R package `stats`.

(iii) if the number of breakpoints estimated in step (i) is larger than 0, the object containing the result from `lm()` is passed to the function `segmented()` from the homonymous package, which performs the iterative procedure shown in Eq. (11).

Then final output contains the updated estimates of the breakpoints and the parameter estimates for the appropriate model between Eqs. (8) and (9). If the original number of threshold temperatures is null, the simple OLS estimate from `lm()` is kept with no further updating.

The methodology presented thus far has a few drawbacks. The relationship between $T$ and $f_s$ is expected to be highly nonlinear, and a segmented straight line may not be able to fully catch this nonlinearity. Furthermore, to ensure that the 350 estimates are physically meaningful, we are forced to use a transformation, in this case $\mathrm{logit}(\cdot)$, to avoid predictions outside the interval of admitted values, [0,1]. This also implies that, once estimates $\hat{f}'_s$ are obtained for the logit-transformed variable, the predicted snowfall must be obtained applying the inverse logit transformation, also known as logistic or expit:

$$\hat{f}_s = \mathrm{logistic}(\hat{f}'_s) = \frac{\exp(\hat{f}'_s)}{\exp(\hat{f}'_s)+1}. \tag{12}$$

The use of invertible transformations in linear regression is common, but it can create issues when back-transforming the 355 predicted value to re-project them onto the scale of the original variable. A typical example is the use of a natural logarithm transformation of the response variable, $y' = \log y$, very common when $y$ takes positive values and follows a positively skewed distribution.

However, if $\log y \approx \beta_0 + \beta_1 x + \varepsilon$, the model in the original scale reads

$$y \approx \beta'_0 e^{\beta_1 x} \varepsilon',$$

with $\beta'_0 = e^{\beta_0}$ and $\varepsilon' = e^{\varepsilon}$. This means that the log-linear model is well specified only if the underlying generating process is characterized by multiplicative errors. Furthermore, assuming a model as in Eq. (7) in the log-linear scale implies that $y' \sim N(\mu, \sigma^2)$ and then that $y \sim LogNorm(\mu, \sigma^2)$, where $LogNorm(,)$ denotes the lognormal distribution. It follows that $E[y'] = \mu$, but $E[y] = \exp(\mu + \sigma^2/2)$: then, by taking the inverse transform of the logarithm of the prediction will give an estimate $\hat{E}(y) = \exp(\hat{\mu})$, while the correct estimator would be $\hat{E}(y) = \exp(\hat{\beta}_0 + \hat{\beta}_1 x + \frac{\hat{\sigma}^2}{2})$, producing a bias given by a factor 365 equal to $\exp(\frac{\hat{\sigma}^2}{2})$.

Similarly, in our setting, the use of the logit transform requires two assumptions to enable us to state that models 7-9 are well specified: (i) that the odds of snowfall against rainfall are log-normally distributed and that (ii) the process on the odds is multiplicative, so that taking their natural logarithm results in a normal distribution and linearizes the model. Taken together, these two assumptions require that the odds ratio $f_s/(1 - f_s)$ follows a logit-normal distribution (Atchison and Shen (1980)). 370 As pointed out by Hinde (2011), the logit-normal assumption implies for $f_s$ a distribution on the interval (0,1) different from the Beta distribution, usually assumed to describe continuous proportion data.


### 3.1.3 Beta regression

While the logit transformation is often useful in practice to project an originally bounded variable onto the real numbers, variables assuming values over a limited interval are often characterized by an irreducible skewness and may manifest het-
eroskedasticity and multimodality as well. In order to account for these features, Ferrari and Cribari-Neto (2004) introduced a regression model based on the beta distribution. In principle, beta random variables take values in (0,1), making it the reference distribution for the statistical modelling of proportions. However, if $y$ is a random variable over any interval $(a, b)$, the transformed variable $\tilde{y} = (y - a)/(b - a)$ assumes values in (0,1) and can, in turn, be modelled through a beta distribution.

We say that $y$ follows a beta distribution with parameters $p$ and $q$, $y \sim B(p, q)$, if its probability density function is

$$f(y; p, q) = \frac{\Gamma(p + q)}{\Gamma(p)\Gamma(q)} y^{p-1}(1 - y)^{q-1}, \qquad 0 < y < 1, \quad p, q > 0 \tag{13}$$

where $\Gamma()$ is the Euler's gamma function. The beta distribution is very versatile and its probability density function can be symmetric or skewed, uniform or U-shaped. Both the parameters appearing in Eq. (13) are shape parameters, $p$ "pulling" the distribution to the left limit, $q$ to the right. As pointed out by Cribari-Neto and Zeileis (2009), this feature is not ideal in a regression framework, since it is difficult to interpret shape parameters in terms of conditional expectations. To circumvent this
issue, it is useful to reparameterize the distribution based on its expected value. If $y \sim B(p, q)$,

$$E(y) = \frac{p}{p + q}, \qquad Var(y) = \frac{pq}{(p + q)^2(p + q + 1)}.$$

Then, it is possible to define two parameters

$$\mu = E(y) = \frac{p}{p + q}, \qquad \phi = p + q$$

so that

$$f(y; \mu, \phi) = \frac{\Gamma(\phi)}{\Gamma(\mu\phi)\Gamma((1 - \mu)\phi)} y^{\mu\phi - 1}(1 - y)^{(1-\mu)\phi - 1}, \qquad 0 < \mu < 1, \quad \phi > 0. \tag{14}$$

In this alternative parameterization, $\mu$ is a location parameter and $\phi$ is a scale parameter. More specifically, $\phi$ is linked to the precision of the distribution, since the larger $\phi^{-1}$ the larger is the variance, which now reads

$$Var(y) = \frac{\mu(1 - \mu)}{1 + \phi}.$$

Let us consider a sample $y_1 \ldots y_n$ drawn from a uni-dimensional random variable $y \sim B(\mu, \phi)$ and $n$ realizations from a set
of $k$ covariates, $x_{i1} \ldots x_{ik}$. The beta regression model reads

$$g(\mu_i) = x_i^T \beta \tag{15}$$

where $\beta = (\beta_1 \ldots \beta_k)^T$ is the vector of regression coefficients, and $g(\cdot) : (0, 1) \mapsto \mathbb{R}$ is a strictly increasing, twice differentiable link function. The main motivation for the inclusion of a link function is to map a bounded variable onto the real line. However, given the possibility to choose among different functions the one that provides the best fit, it also adds flexibility to the model





(see, for example, Cribari-Neto and Zeileis (2009) for a list of suitable link functions for beta regression). The simplest link for
the beta regression is the logit transform already mentioned in Section 3.1.2, which we will adopt in our analysis.

In operational terms, the beta regression described above is implemented in the R package `betareg`, described by Cribari-
Neto and Zeileis (2009). This package does not implement the regression in such a way that outputs can be passed to the
package `segmented` for segmented regression. Then, to perform segmented beta regression, we will use the thresholds
defined through the breakpoint search, running the model estimation separately for each of the regimes found in the specific
grid point using `lm()`.

### 3.1.4 Spline regression

At the beginning of Section 3.1, we have mentioned that we exclude the use of nonlinear S-shape function fitting due to several
co-existing reasons, which we recall here. First, it is possible that there exist areas where the relationship between DJF snow
fraction and temperature is not well described by an inverse S-shape: for instance, high altitude locations in the Alps, or areas
of Scandinavia are very unlikely to receive high amounts of rain during the winter, making it reasonable to assume that the
right branch of the inverse S-shape would not be observed. Moreover, different locations may exhibit shapes that are better
described by different S-shaped functions, making this practice suitable for studies based on a single time series or on data
from a relatively small region, but not on a large grid such as the EURO-CORDEX domain. Finally, NLS estimation depends
on the chosen optimization algorithm and, most importantly, on the initial values specified for the parameters: these should be
reasonably close to the true values, which are, in turn, unknown.

On the other hand, the segmented (both logit-linear and the beta) regressions previously described also present some draw-
backs. First, a piecewise linear function can only be a coarse approximation of the underlying nonlinear relationship. Second,
segmented regression does not guarantee that the regression lines in different regimes match at the threshold points. In order
to fit functions that are more flexible in shape and smooth at the threshold temperatures, we rely on splines. Spline regression
can be thought of as an improvement of traditional power transform and polynomial regression, where the regression splines
are piecewise polynomials constrained to meet smoothly at the knots, i.e. the transition $x$-points, in our case the threshold tem-
peratures. One main drawback of polynomials is that they display unpredictable tail behavior, making polynomial regression
a poor methodology to model data with tail asymptotic behavior, or where extrapolation may be needed to predict $y$ from $x$
values that are not observed in the training set. Moreover, polynomial regression is highly nonlocal, meaning that features of
the data over small regions of the $x$ domain may heavily bias the overall model.

Let us consider the general case of a problem with $k = 1, \ldots, K$ knots $\xi_k$. We call a spline of order $M$ a piecewise polynomial
of order $M - 1$ with continuous derivatives up to the order $M - 2$. Common choices are $M = 1$, corresponding to the case
of a piecewise constant function, $M = 2$ (continuous piecewise linear function) and $M = 4$ (cubic spline). Notice that the
case $M = 2$ is analogous to a segmented regression with matching regression lines. Splines of order $M$ with $K$ knots have





$D_f = M(K+1) - K(M-1)$ degrees of freedom, and can be decomposed on a basis of $D_f$ functions $h(\cdot)$:

$$y = \sum_{m=1}^{D_f} \beta_m h_m(x) + \varepsilon$$

$$h_k(x) = x^{k-1}, \qquad\qquad k = 1, \ldots, M,$$

$$h_{k+M}(x) = (x - \xi_k)_+^{M-1}, \qquad k = 1, \ldots, K.$$

(16)

In our case, we consider as possible values of the knot number $K = 1, 2$, corresponding to the one or two threshold temperature scenarios. For a more comprehensive overview on spline regression, see e.g. Fox (2015) and Harrell Jr (2015).

## 3.2 Design of the experiment and model validation

We assess the performance of each model in recovering $f_s$ as a function of $T$ and then $SF = f_s \cdot P_{tot}$ using a test set from ERA5 for model validation. Then, we select the best model as the one providing the out-of-sample best prediction of snowfall in ERA5, and we use parameter estimates to approximate $SF$ in the IPSL_WRF model, assessing its performance in terms of bias correction with respect to the original model output. In the following, the apexes $E5$ and $IW$ denote ERA5 and IPSL_WRF variables, respectively. Moreover, the hat superscript denotes estimations obtained from the regression models. For example, $P_{tot}^{E5}$ denotes the total precipitation in ERA5, $\hat{f}_s^{IW}$ denotes the snow fraction estimated from $T$ in the IPSL_WRF climate simulations. If no apex is shown, we refer to the indicated variable in general. Furthermore, as already mentioned, $T_{(low,high)}^*$ denotes the threshold temperature(s) obtained with the breakpoint analysis, and $\tilde{T}_{(low,high)}^*$ their updated version in the case of segmented logit regression. In all the models including thresholds we use the results of the breakpoint analysis $T^*$, except for the segmented logit regression, where we use the updated version $\tilde{T}^*$. Finally, we use $\theta$ to denote the generic parameter vector of each model including $\beta$ and the temperature thresholds, and $\hat{\theta}$ the corresponding estimate.

As a preparatory step, we transform the data so that it is included in (0,1) without assuming the boundary values. In fact, neither the beta regression procedure, nor the segmented or simple regression on logit-transformed data are well behaved in case the variable assumes the limiting values 0 and 1. To circumvent this problem, Smithson and Verkuilen (2006) propose a transformation to effectively shrink the interval:

$$f_s \rightarrow \frac{f_s(n-1) + 0.5}{n},$$

(17)

where $n$ is the sample size. Notice that the interval amplitude will depend on $n$: for example, for $n = 50$ the interval reduces to (0.01,0.99), while for $n = 100$ it ranges in (0.005,0.995). In the following, all variables are transformed using Eq. (17) for all the models, except for the naive threshold estimation. Since we consider this adjustment as part of the regression procedure, no different notation will be used to denote the transformed variables.

For the model selection and validation steps, we use the entire 1979-2018 available period. For the STM model, we choose $T^* = 2°C$ as the single threshold, following Faranda (2020). Then, we put $\hat{f}_s^{E5} = 1$ if $T < T^*$ and $\hat{f}_s^{E5} = 0$ if $T > T^*$. To estimate snow fraction models based on regression, for each point in the ERA5 grid, if the total number of snowfall events is a least $n = 30$, we randomly select half of the values as a training set and the remaining is used as a test set; otherwise, the





grid point is excluded from the analysis. Each model is evaluated on the training set to obtain the parameter estimate $\hat{\theta}$ (with the exception of the STM). Then, $\hat{\theta}$ is used to estimate $\hat{f}_s^{E5}$ in the test set, and we finally obtain the estimated snowfall as $\widehat{SF}^{E5} = \hat{f}_s^{E5} \cdot P_{tot}^{E5}$.

The performance of the model is assessed comparing true and predicted $SF$ values in the test set. In particular, for each grid point we compute two error measures, the mean absolute error (MAE) and the root mean square error (RMSE):


$$
MAE = \frac{2}{n} \sum_{j=1}^{n/2} |\widehat{SF}_j^{E5} - SF_j^{E5}|
$$

$$
RMSE = \sqrt{\frac{2}{n} \sum_{j=1}^{n/2} (\widehat{SF}_j^{E5} - SF_j^{E5})^2}
$$

(18)

We do not base our model selection on information criteria (IC), such as the Akaike or the Bayesian IC, mainly for practical reasons. In fact, neither the naive STM nor the piecewise beta regression are associated to an IC, the former because it is not in fact a parametric model, the latter because, due to software limitations, we perform the regression segments separately, so that it is not possible to recover one IC value. However, this is not a concern, since our focus is on prediction, rather than on
the interpretation of model results.

We select the best model as the one giving the best performance in terms of minimum $MAE$ and $RMSE$ on the widest possible part of the considered domain. Once the best model is found, we repeat the estimation for each grid point using the entire sample size $n$, instead of dividing it in training and test set. Then, we use the resulting $\hat{\theta}$ and $T^{IW}$ to estimate the snow fraction in the climate model, $\hat{f}_s^{IW}$, and the corresponding snowfall $\widehat{SF}^{IW} = \hat{f}_s^{IW} \cdot P_{tot}^{IW}$. Then, we compare both $SF^{IW}$ and
$\widehat{SF}^{IW}$ to $SF^{E5}$, and we assess whether the version estimated with the chosen model and using bias adjusted total precipitation is closer to reanalysis than the raw model output, showing that the statistical model can be used to replace BC for snowfall.

The capability of the chosen model(s) to perform a BC-like task can be evaluated in terms of similarity between the distribution of the modelled daily snowfall and the reanalysis values. We will use three measures of dissimilarity between the distributions: the Kolmogorov-Smirnov (KS) statistics, the Kullback-Leibler (KL) divergence and the $\chi^2$ divergence. The KS
statistics was originally designed to test the null hypothesis that two samples are drawn from populations with the same probability distribution (Darling, 1957). This test, compared to others previously designed, shows a higher power, i.e. a better ability to correctly reject the null hypothesis when it is false. However, when samples to be compared are large, the KS test rejects the null hypothesis even in presence of very small differences between the distributions, which we realistically expect to observe even in case of what we could consider a good performance of our models. For this reason, rather than performing the test, we
use the KS statistics as a measure of the proximity between reanalysis and model distributions. In general, given two samples $x$ and $y$ of size $n$ and $m$ from two random variables $X, Y$ with distribution functions $F(x), G(y)$, the KS statistics is constructed as

$$
D_{n,m}(x,y) = \sup |F_n(x) - G_m(y)|,
$$

where $F_n(x), G_m(y)$ are the empirical cumulative distribution functions (ECDF) of the two samples, obtained as $F_n(x) =$
$\frac{1}{n} \sum_{i=1}^{n} I_{[-\inf, x]}(x_i)$ and in the same way for $G_m(y)$.





While the KS statistic can be seen as a distance, since $D_{n,m}(x,y) = D_{m,n}(y,x)$, the same does not apply to the KL and the $\chi^2$ measures, that are usually referred to as divergences. Both can be computed between two samples, one of which is considered as drawn from a reference distribution: depending on which sample is taken as the reference, the value of the $\chi^2$ and of the KL changes. For a comprehensive review of the role of divergences and distances in statistical inference, see Basu et al. (2011) and Pardo (2018). In presence of two samples $x$ and $y$ as described above, we consider $y$ to be drawn from the reference distribution. Let $f(x), g(y)$ be the probability density functions of the two variables and $f_n(x), g_m(y)$ their empirical version, i.e. their histogram or a kernel density estimation over a common set of bins $h = 1, \ldots, H$. The $\chi^2$ divergence reads

$$\chi^2(x,y) = \sum_{h=1}^{H} \frac{(f_n(x)_h - g_m(y)_h)^2}{g_m(y)_h}$$

and the KL divergence can be obtained as

$$KL(x,y) = \sum_{h=1}^{H} \log\left(\frac{f_n(x)_h}{g_m(y)_h}\right) f_n(x)_h.$$

Notice that both quantities are not defined if the empirical reference distribution in the $h$-th bin is zero. Given the high positive skewness of the observable of interest in this paper, problems may arise in the right tail of the distribution, if there are zero-valued density estimates in $g_m(y)$ at bins where $f_n(y)_h > 0$. Since we are using these two quantities to evaluate the overall forecasting performance of the models, rather than to draw strong theoretical conclusions, we simply circumvent this problem by cutting the right tail after the first bin with $g_m(y) = 0$ during the computation.

## 4 Results

### 4.1 Threshold temperatures

The results of the breakpoint analysis for the search of optimal threshold temperatures, described in Section 3.1.2, are shown in figure 2. For each grid point, we run the search algorithm saving the resulting breaking point positions, and we plot the corresponding temperature values. We discard all grid points where the total number of DJF snowfall events is smaller than 30, which is the cut-off chosen as minimal sample size for the regressions.

First of all, it is worth noticing that at least one threshold temperature is found for each grid point, even though having no breakpoint is an admissible outcome from the search algorithm. This corroborates the idea that some sort of transition between two regimes is to be expected concerning the relationship between $T$ and $f_s$. Panel (a) in Fig. 2 displays the locations and temperature values where only one threshold is found. These are also likely locations where the choice of a STM may produce good results. However, it is clear from the figure that the values of the threshold are quite variable and, as opposite to the usual choice for a STM framework, are mostly negative. In particular, strongly negative values ($-5°C \lesssim T^* \lesssim -10°C$) are found over the Alps and Scandinavia with the exception of the Central-Northern parts of Norway; milder but still markedly negative values are also observed over parts of Eastern Turkey, Austria, Czech Republic, Slovakia, Hungary, Bulgaria, Romania and Moldova. All these areas are characterized by a large or extremely large number of snowfall events (Fig. 1, panel (b)), but





only the Alps and Eastern Turkey are also areas with a large amount of total snowfall. The other two regions characterized by extreme total snowfall amounts, Iceland and the coast of Norway with the exception of the most South-Western part, are instead not at all represented by this single threshold configuration. Weakly negative thresholds are also found over Western France and Southern Germany, while weakly positive single thresholds are found only for the UK, Spain, Southern France,
peninsular Italy and, with more markedly positive values ($T^* \sim 5°$C), the Northern parts of Morocco, Algeria and Tunisia.

Panels (b) and (c) in Fig. 1 show, respectively, $T^*_{low}$ and $T^*_{high}$ for all the remaining grid points, characterized by a double threshold. These areas include Iceland and Central-Eastern Europe at all latitudes, excluding the aforementioned areas characterized by single thresholds. Concerning $T^*_{low}$, markedly negative values ($-5°$C $\lesssim T^*_{low} \lesssim -10°$C) are found over Iceland and the most snowy parts of Norway. Over the rest of the interested area, values are negative everywhere, with a clear gradient
moving from South-West to North-East, reaching $-10°$C $\lesssim T^*_{low} \lesssim -15°$C over Sweden, Southern Finland and Russia. Much less uniformity is observed for the upper threshold as large areas display negative values also for $T^*_{high}$. Iceland, Scandinavia and Russia display markedly negative values, up to ($T^*_{high} \sim -5°$C), while the rest of the continent, including Western Turkey, is characterized by positive values, mostly below 5°C and an imaginary line of $T^*_{high} \sim 0°$C going from Estonia to Ukraine. It is worth to remark that we refer to daily temperatures, and it is therefore quite surprising to observe that there exists a
non-trivial relationship between $T$ and $f_s$ even when both thresholds are below zero.

**Threshold temperatures adjusted for segmented regression**

As illustrated in detail in Section 3.1.2, when the relationship between $T$ and $f_s$ is modelled using segmented logit-linear regression, the threshold temperatures are adjusted according to Eq. (11), producing new values $\tilde{T}^*$. The results of this adjustment are shown in Fig. 3.
Compared to the thresholds $T^*$ obtained from the breakpoint analysis (Fig. 2), $\tilde{T}^*$ show several differences, some of which are rather counter-intuitive. The spatial distribution of negative and positive values of single threshold temperatures are essentially inverted, with negative values over most of Southern and Western Europe, and positive values over the Alps, East Turkey and Scandinavia (Fig. 3 (a)). Lower thresholds (Fig. 3 (b)) are mostly negative, except over Russia and parts of Scandinavia; values are, however, very close to $\tilde{T}^*_{low} \sim 0°$C, while the results of the breakpoint analysis show a clear South-West to North-
East gradient, with very low temperatures over Russia. Upper threshold values (Fig. 3 (c)) show overall positive temperatures, with higher values over Iceland, Norway, Russia and Easter Europe.

We find that such results are less interpretable in terms of meteorological processes, compared to the threshold temperatures estimated via breakpoint analysis. For this reason, we only adopt thresholds $\tilde{T}^*$, $\tilde{T}^*_{low}$ and $\tilde{T}^*_{high}$ for the logit-linear segmented regression, while we use the breakpoint analysis results as thresholds for the piecewise beta regression and for the spline
regression.

### 4.2   Model selection

We select the best performing model in terms of snowfall prediction as described in Section 3.2: for every grid point, we split the time series of snowfalls in a training and a test set, we estimate each model in the training set, and we use parameter




estimates to predict the values in the test set. In Fig. 4 we show a summary of the results in terms of the chosen error measures:
each boxplot refers to the values of the error measure over the entire domain for grid points with more than 30 snowfall events.
The tested models include the naive STM, both the logit-linear and the beta regression performed without any segmentation
respect to the temperature, the segmented logit-linear and the cubic spline regressions.

We do not show results for the linear splines, as they are virtually identical to the cubic spline case, which we prefer due to a
potentially greater flexibility and capability to extrapolate to unobserved values in the prediction phase. Moreover, we exclude
the segmented beta regression due to its poor stability: the number of points over which the model does not converge is 44783
over a grid of 48321 points. For comparison, the number of points where the model estimation is null is 24316 for the naive
STM (due to sea masking and exclusion of points with less than 30 snowfalls), 24469 for the cubic spline, 25077 for the simple
logit regression, 25387 for the segmented regression and 25743 for the simple beta regression. Of these, the 24316 of the naive
STM model are due to masking out the ocean and grid points with less that 30 snowfall events. We do not display outliers in
the boxplot for greater graphical clarity.

We show results for both the snow fraction, directly derived from the model forecasting (Fig. 4 (a)), and the snowfall,
obtained as $\widehat{SF}^{E5} = \widehat{f_s}^{E5} \cdot P_{tot}^{E5}$ (Fig. 4 (b)). Results for the two variables are qualitatively coherent across error measures and
across variables, even though the distribution of both error measures are quite symmetric for the snow fraction, while positively
skewed for estimated snowfall.

From a visual inspection, it is clear that the two regressions without thresholds provide the poorest performances, followed
by the naive STM. The latter, however, is characterized by much less positive outliers. This is not surprising, because assuming
a binary classification is a gross approximation in case of mixed precipitation, but it is be accurate in case of precipitation
happening at markedly positive or negative temperatures, avoiding large errors in case of extreme events. The segmented
regression and the cubic splines produce the best performances.

In order choose the best possible methodology based on quantitative considerations, we test for significant differences among
groups using the rank test proposed by Kruskal and Wallis (1952) as a nonparametric alternative to the classic ANOVA. In fact,
conditions to apply a one-way analysis of variance (ANOVA) are not met, since the distribution of the MAE and RMSE
are clearly non homoskedastic across models, and visibly non-Gaussian at least in the case of the snowfall. To compute the
Kruskal-Wallis $H$ test statistic, all observations must be ranked together regardless of the group to which they belong. If there
are no ties, i.e. no identical observations, the test statistics reads

$$H = \frac{12}{N(N+1)} \sum_{g=1}^{G} \frac{R_g^2}{n_g} - 3(N+1) \tag{19}$$

where $G = 5$ is the number of groups, defined by the five chosen models; $n_g$ is the sample size and $R_g$ is the sum of the ranks
of group $g$, with $N = \sum n_g$. In case of ties, each tied observation is assigned the mean of the ranks for which it is tied, and
then the test statistics is corrected:

$$H \rightarrow \frac{H}{1 - \frac{\sum_g T_g}{N^3 - N}} \tag{20}$$





with $T_g = t_g^3 - t_g$, where $t_g$ is the number of ties in group $g$. If $H$ is large, the null hypothesis of no significant differences among groups is rejected at the chosen level. The distribution of $H$ is tabulated, and if the sample sizes in the groups are large enough, it is approximately distributed as a $\chi^2$ random variable.

We perform a total of four Kruskal-Wallis tests, on snow fraction RMSE, snow fraction MAE, snowfall RMSE, and snowfall MAE. In all cases, the null hypothesis must be rejected, with virtually null p-values (all $< 2.2 \cdot 10^{-16}$, which is the smallest value resolved in R). From this, we can conclude that at least one model produces results that display stochastic dominance respect to at least another model. However, we cannot establish exactly which groups are concerned.

     To this purpose, we rely on post-hoc testing using the pairwise Wilcoxon rank sum test (Wilcoxon, 1992), a nonparametric alternative to pairwise Student's $t$ tests suited for non-Gaussian samples, also known as Mann-Whitney $U$ test. The procedure

consists of the following steps: i) two samples are merged and ranked; ii) the sum of the ranks is computed separately for data belonging to sample $A$ and $B$, and denoted $R_A$ and $R_B$ respectively; iii) the Mann-Whitney $U$ test statistic is obtained as $U_A = R_A - n_A(n_A + 1)/2$. In case of ties, the rank in the midpoint between the two closest non-tied ranks is used. Under the null hypothesis, $U$ follows a known tabulated distribution, which converges to a Normal distribution for large enough samples ($n \gtrsim 20$). Since for $G$ groups we perform $G(G-1)/2$ tests, the p-values must be corrected to control the familywise error rate

(FWER). We choose the method introduced by Benjamini and Hochberg (1995), which is usually recommended because it provides higher power than other common FWER correction methods. However, we also compare results to the ones obtained using the simple Bonferroni correction (Bonferroni, 1936), which turn out to be equivalent in terms of statistical significance. Results point to significant differences between all possible pairs of groups. However, due to the very large sample size, the tests may be extremely sensitive even to very small differences, possibly leading to a high probability of rejection even with

differences that are unrecognizable in practice.

     The close similarity of results from the segmented and the spline regression is also evident from the summary statistics of the distributions of the two error measures for the two variables, shown in Table 1. These two models are in close competition, with the segmented regression overall slightly better than the cubic splines, except for the interquartile range (IQR) of both indicators for snow fraction, and other statistics that are identical at least to the considered precision (both standard deviations

for snow fraction and median for snowfall MAE).





| Snow Fraction | | | | | | | | |
|---|---|---|---|---|---|---|---|---|
| | RMSE | | | | MAE | | | |
| Model | Mean | St. Dev. | Median | IQR | Mean | St. Dev. | Median | IQR |
| naive | 0.192 | 0.0862 | 0.203 | 0.140 | 0.315 | 0.0991 | 0.337 | 0.149 |
| logit-linear | 0.390 | 0.123 | 0.385 | 0.207 | 0.474 | 0.117 | 0.479 | 0.197 |
| beta reg. | 0.325 | 0.0866 | 0.330 | 0.139 | 0.400 | 0.0878 | 0.410 | 0.145 |
| segmented reg. | 0.159 | 0.105 | 0.148 | 0.0944 | 0.242 | 0.101 | 0.237 | 0.0886 |
| cubic spline | 0.163 | 0.105 | 0.152 | 0.0943 | 0.247 | 0.101 | 0.239 | 0.0858 |

| Snowfall | | | | | | | | |
|---|---|---|---|---|---|---|---|---|
| | RMSE ($10^{-4}$ m) | | | | MAE ($10^{-4}$ m) | | | |
| Model | Mean | St. Dev. | Median | IQR | Mean | St. Dev. | Median | IQR |
| naive | 0.184 | 0.151 | 0.156 | 0.154 | 0.530 | 0.337 | 0.474 | 0.356 |
| logit-linear | 0.493 | 0.316 | 0.441 | 0.294 | 0.896 | 0.490 | 0.804 | 0.419 |
| beta reg. | 0.403 | 0.249 | 0.359 | 0.207 | 0.739 | 0.402 | 0.655 | 0.329 |
| segmented reg. | 0.172 | 0.277 | 0.110 | 0.121 | 0.419 | 0.447 | 0.322 | 0.273 |
| cubic spline | 0.178 | 0.279 | 0.113 | 0.124 | 0.436 | 0.461 | 0.322 | 0.285 |

**Table 1.** Summary statistics of the distributions of the model RMSE and MAE for snow fraction and snowfall: mean, standard deviations, median and interquartile range.





| Correlation coefficient | | | | |
|---|---|---|---|---|
| Model | Mean | St. Dev. | Median | IQR |
| naive | 0.473 | 0.147 | 0.487 | 0.194 |
| logit-linear | 0.280 | 0.340 | 0.390 | 0.174 |
| beta reg. | 0.309 | 0.349 | 0.422 | 0.173 |
| segmented reg. | 0.622 | 0.164 | 0.670 | 0.157 |
| cubic spline | 0.595 | 0.200 | 0.661 | 0.195 |

**Table 2.** Summary statistics of the distributions of correlation between true (reanalysis) and predicted snow fraction. 'IQR' indicates the amplitude of the inter-quartile range.

As a further criterion to choose the best performing model, we consider the Pearson's correlation coefficient computed, at each grid point, between the snow fraction observed in ERA5 and predicted using the five statistical models under investigation. Notice that the factor linking $f_s$ and $SF$ is $P_{tot}^{E5}$ for both true reanalysis and reconstructed data, so that results in terms of correlation coefficient are exactly the same for both variables. The boxplots, shown in Fig. 5 (a) and constructed as described

for MAE and RMSE, show a similar behavior, with the segmented regression and the cubic spline regression performing sensibly better than the other three models. As in the previous case, the Kruskal-Wallis test indicates a significant stochastic dominance of at least one group on the others, but the pairwise Mann-Whitney test suggests significant differences between all possible pairs. Both the visual inspection of the boxplots and the summary statistics shown in Table 2, hint that there may be a slight preference towards the segmented regression, which has the highest mean and median, and the lowest variance and IQR.

Nevertheless, it must be noticed that the spline regression algorithm converged for 918 more grid points than the segmented regression: not only this could slightly skew the statistics, but the capability of the model to represent the highest possible number of grid points is also a criterion to consider to choose the most useful methodology. Overall, the performance of these two models over the EURO-CORDEX domain (see Fig. 5 (b) and (c) for a comparison) is mostly the same.

It is worth noticing how areas characterized by frequent and abundant snowfall, such as the Alps, Scandinavia, East Turkey,

correspond to the lowest values of the correlation coefficient. This is also true when considering, for example, the naive STM model (Fig. 5 (d)) which, however, displays overall sensibly lower values of the correlation coefficient.

It is now clear that the segmented regression and the spline regression perform significantly better at reconstructing the snowfall compared to the naive STM and to simpler regressions with no threshold temperatures. However, these results do not constitute a strong evidence towards a better performance in reconstructing snowfall in practical cases, i.e. when it is

unobserved or severely biased. To this purpose, we apply both methods to reconstruct the snowfall in a climate projection model and we assess which one produces the least biased snowfall using bias adjusted temperature and precipitation as an input.




### 4.3 Bias correction of climate simulations

We now assess the performance of both the segmented logit regression and the cubic spline regression on the output of the IPSL-WRF climate simulations for the period 1979-2005. For this model, we have near-surface temperature and total daily precipitation bias corrected using CDF-t with respect to ERA5. However, available snowfall data is not corrected, and presents non negligible differences compared to ERA5 in terms of both long run statistics and probability distribution of daily snowfall. Let us denote $\widehat{SF}^{LR}$ and $\widehat{SF}^{SP}$ the estimated snowfall obtained with logit segmented regression and cubic splines respectively; moreover, let us denote $\Sigma SF$ the total cumulated snowfall over the period 1979-2005. Fig. 6 shows the differences $\Delta^{IW} = \Sigma SF^{IW} - \Sigma SF^{E5}$ (a), $\Delta^{LR} = \Sigma \widehat{SF}^{LR} - \Sigma SF^{E5}$) (b) and $\Delta^{SP} = \Sigma \widehat{SF}^{SP} - \Sigma SF^{E5}$ (c). The IPSL-WRF model displays several areas of positive bias, especially over mountain areas such as the Alps, Massif Central, Pyrenees, Carpathians, Turkey and Georgia, plus Southern Norway. Values in these areas are mostly $5 \lesssim \Delta^{IW} \lesssim 10$ m, with peaks up to $\Delta^{IW} \sim 20 - 30$ m over the Alps and Georgia, values that are comparable to the total amount of snowfall observed in ERA5 (i.e. with relative biases close to 100%). A strong negative bias area is instead observed over the coast of Norway, with absolute values around 15 m, once again close to 100% of $\Sigma SF^{E5}$ (see Fig. 1 (a)). A more heterogeneous bias is observed on Iceland, with negative values on the Western and negative values on the Eastern side of the country, both with a magnitude of $\Delta^{IW} \sim 5$ m.

Panels (b) and (c) in Fig. 6 show that, overall, $\Delta^{LR}, \Delta^{SP} < \Delta^{IW}$, with a strong reduction of the bias especially over the most critical areas including mountain ranges and Norway. We will give special attention to these two areas in the next Section 4.3.1. To complete the evaluation of the performance over the entire EURO-CORDEX domain, we must consider that preserving the total amount of snowfall over a long period may not be enough, if the aim of the snowfall estimation is to conduct hydrological studies or climatological analysis focusing on extremes, seasonality or other features that require a finer resolution. This is the reason why, usually, bias correction is not performed only on the mean level of the observable, but in such a way to correct as much as possible the entire probability distribution.

We evaluate the performance of the logit segmented and of the cubic spline regression, and compare it to the values produced by the IPSL-WRF model, using three measures of proximity between probability distributions as described in Section 3.2 (KS, KL, $\chi^2$): at each grid point, we estimate the three divergence measures between the empirical probability distribution of ERA5 and of each model. The results are summarised in Fig. 7 (a) through the boxplots of the values considering the entire domain. Outliers are excluded for greater graphical clarity. For each model, the three divergence measures assume overall similar values. In terms of model performance, the unadjusted snowfall from the IPSL-WRF simulation is clearly characterized by a larger distance between the distributions compared to estimates obtained using either of the regression models under examination. The logit segmented regression appears to be the best model in terms of both median and variability, consistently displaying the smallest IQR for all the three measures. However, while $\chi^2$ and KL produce very homogeneous values over the domain (not shown), the KS is way more spatially differentiated and also has larger outlier values. Panels (b) and (c) in Fig. 7 show the spatial distribution of the difference between IPSL-WRF KS distance compared to segmented logit and cubic spline regression, respectively. The patterns and values are very similar, and differ sensibly only over a small number of points. There is a clear spatial coherence, showing that improvements in terms of KS statistics concern Eastern Europe, Scandinavia and the already





mentioned snowy areas characterized by large biases, while its values for the regression models are sensibily higher over UK, France, Benelux, Italy, North Africa and Aegean Sea. Considering once again the domain as a whole, the KS distance for the IPSL-WRF model has a larger mean and wider IQR compared to the regression models, while these display a strong positive

skewness with a long right tail. Indications from the three divergence measures overall point to a better performance of the regression models, although without a clear outperformance of either of them.

### 4.3.1    Regional extremes: Alps and Norway

The results discussed so far show that the IPSL-WRF model is affected by a spatially inhomogeneous and sometimes important bias, and that the snowfall reconstruction based on the two models selected in Section 4.2 produces more realistic values on the

ensemble of the grid points. In particular, the greatest advantage seems to be in the most sensitive areas, such as the Alps and Norway, where post-correction biases are comparable to the rest of the domain, while in the IPSL-WRF model are roughly up to an order of magnitude larger. In this paragraph, we focus in particular over these two areas. We define the Alps region based on Eurostat 2016 NUT3 statistical units, choosing provinces belonging to France, Switzerland, Italy and Austria characterized by large values of total 1979-2005 snowfall and large differences between ERA5 and IPSL_WRF. In particular, we include the

regions marked by the following list of NUTS3 codes:

| France | Switzerland | Italy | Austria |
|--------|-------------|-------|---------|
| FRK27  | CH011       | ITC11 | AT322   |
| FRK28  | CH012       | ITC12 | AT323   |
| FRL01  | CH013       | ITC13 | AT331   |
| FRL02  | CH051       | ITC14 | AT332   |
| FRL03  | CH053       | ITC16 | AT333   |
|        | CH054       | ITC20 | AT334   |
|        | CH055       | ITC41 | AT335   |
|        | CH056       | ITC42 | AT341   |
|        | CH062       | ITC43 | AT342   |
|        | CH063       | ITC44 |         |
|        | CH064       | ITH10 |         |
|        | CH065       | ITH33 |         |
|        | CH070       |       |         |

**Table 3.** NUTS3 units used for the definition of the Alps region, divided by country.

The IPSL-WRF simulation is characterized by a strong positive bias over the Alps, with values up tp $\sim 30$ m difference in total 1979-2005 snowfall, as shown in Fig. 8 (a). Panels (b) and (c) of Fig. 8 clearly show the extreme reduction of the bias for both the logit segmented and the cubic spline regression, with values that pass from $\sim 20 - 30$ m to $\sim 1$ m especially in the most affected areas of Central-Western Alps and Extreme West Austria. Overall, the median bias remains slightly positive,

but the IQR of the differences with respect to ERA5 is reduced by around an order of magnitude, as evident from Fig. 8 (d). Since comparing IPSL-WRF and its adjusted versions to ERA5 does not provide a one-on-one correspondence between





snowfall events, it is not possible to compute correlation coefficients between reanalysis and model snowfall time series at each grid point as in Fig. 5. Nevertheless, it is still possible to compute the correlation between grid-point total snowfall in each model and ERA5 considering the region as homogeneous. Fig. 9 (a) shows the scatterplots of snowfall simulated in the

IPSL-WRF model and reconstructed using segmented logit and cubic spline regression, in comparison to the reference line with null intercept and unit slope. The left column Table 4 reports values of the intercept and slope of the least squares line, plus the coefficient of determination (which is simply the square of the correlation coefficients added to the scatterplot); the asterisk marks coefficients that are significantly different from 0 at the 5% level. The slope estimate is accompanied by its 95% confidence interval, obtained as $\pm 1.96\,s.e.$, where $s.e.$ is the standard error. A model providing ideal results would display a

non significant intercept, a significant slope close to 1 (i.e. with 1 included in the confidence interval), and an $R^2$ as close to 1 as possible.

The least squares line for the IPSL-WRF has a significantly negative intercept, a slope significantly different from 1, and a determination coefficient $R^2 = 0.73$. Conversely, the two regression models produce statistically identical results, with a null intercept, a slope including 1 at the bottom of the confidence interval, and an increased $R^2 = 0.97$. These results demonstrate

a marked improvement of the long run snowfall statistics over the Alps obtained with segmented or cubic spline regression modelling, compared to the uncorrected climate model snowfall. However, as already stressed, bias adjustment usually aims at correcting as much of the distribution of the observable as possible. To assess if and how well our models work in this sense, we compare the ECDF of daily snowfall, once again considering the region as homogeneous. Fig. 9 (b) shows the comparison; the y axis has been cut between 0.5 and 1 to show with better clarity the portion of the distribution with the largest differences.

Once again, the similarity between the results of the two regression models is such that it is impossible to distinguish between the two curves, while the IPSL-WRF distribution is sensibly different. In particular, not only there is a quite large shift in the location of the distribution, but the tails appear to be much better represented by the regression models, pointing to a higher capability to catch extreme snowfall events.

We repeat the same analysis for Norway, another sensible area where snowfall is an important source of precipitation and

the climate model shows large biases. As shown by Fig. 10 (a), IPSL-WRF presents a strong negative bias, up to -10 m of total 1979-2005 snowfall, on the entire Western and Northern sides of the country, and positive biases up to $\sim 5$ m on the Southern areas, making snowfall over one of the most snowy areas in Europe heavily misrepresented. Again, Fig. 10 (b), (c) show the reconstructed snowfall using segmented logit and cubic spline regression, respectively. In this case, the bias pattern is still present after correction, with the exception of a passage from positive to negative values over the Southern coast. However,

the magnitude of these residual bias is reduced approximately by a factor 5 compared to IPSL-WRF. As for the Alps, the boxplot of total snowfall over the country, shown in Fig. 10 (d), show a very similar performance by the two regression models, with a strong reduction compared to the climate simulations. Concerning the link between simulated/predicted and reanalysis cumulated snowfall data, for IPSL-WRF this is very poorly represented by a linear relationship, as visually shown in Fig. 11 (a). The least squares straight line has a significantly positive intercept and a significant slope with an effect size much smaller

than the one corresponding to the perfect prediction; correlation is also much lower than in the Alps case, with only 12% of the variance explained by a linear relationship (see Table 4). Again, regression models produce a sizeable improvement in snowfall



estimation, with $R^2 = 0.98$ for both techniques. However, the segmented logit displays a very small but statistically significant negative intercept, and both slopes are slightly smaller than 1, even though much closer, with values around 0.95 as opposed to the 0.25 of the IPSL-WRF. The improvement is clear also looking at the ECDF of daily snowfall over the region (Fig. 11 (b)), even though, in this case, it is also possible to distinguish a slight difference between the segmented and the cubic spline regression, with the latter performing slightly better.

In summary, we have shown that by combining a breakpoint search algorithm and a flexible regression method - be it a segmented linear or a cubic spline regression - allows for a reliable snowfall reconstruction in climate simulations. This method proves effective both in correcting heavy mean biases and in preserving the shape of the entire probability distribution of (daily) snowfall, rather than only long run totals. This result is crucial for studying the characteristics of future snowfall in a wide range of environments, encompassing both regions characterised by frequent and abundant snowfall in cold climates and temperate areas where occasional snowstorms and heavy wet-snow events can cause serious loss and damage.

|  | Alps | | | Norway | | |
|---|---|---|---|---|---|---|
| Model | Intercept | Slope | $R^2$ | Intercept | Slope | $R^2$ |
| IPSL-WRF | -5.06* | $3.08 \pm 0.25$* | 0.73 | 3.13* | $0.25 \pm 0.045$* | 0.12 |
| logit-linear | 0.05 | $1.02 \pm 0.02$* | 0.97 | -0.09* | $0.94 \pm 0.009$* | 0.98 |
| cubic spline | 0.05 | $1.02 \pm 0.02$* | 0.97 | -0.07 | $0.95 \pm 0.009$* | 0.98 |

**Table 4.** Summary statistics of the linear relationship between modelled and reanalysis snowfall over the Alps and Norway.

## 5 Conclusion

We have presented two statistical methods equally effective in estimating the snowfall fraction of total precipitation, provided that a reliable measure of near-surface temperature is available. This is a relevant problem in both hydrology and climatology, since an accurate estimation of snowfall is a troubled objective in case of both observed or simulated precipitation.

In case of observational data, especially over large areas where a single weather station is not representative, snowfall is often unobserved due to difficulties in making its measurement an automated procedure. On the other hand, climate model outputs often include snowfall, but this is affected by a bias arising from the physical and mathematical approximation contained in the model scheme. For other variables such as temperature and total precipitation, we can rely on well established and relatively simple univariate bias correction methods, that can be applied pointwise in case of gridded data. However, snowfall presents more challenges, since not only the magnitude but also the number of events is biased as a consequence of biases in the temperature. Thus, its correction would require conditioning on temperature and precipitation, and possibly to include a stochastic generator of snowfall events to correct snowfall frequency, other than snowfall magnitude. Therefore, the availability of simple methods to reconstruct snowfall as the snow fraction of total precipitation is a great advantage in contexts where much





more complex and computationally costly procedures should be created and applied in order to obtain an accurate snowfall measurement.

The techniques applied in the existing literature mainly consist of a binary representation based on a threshold temperature, linear interpolations between two thresholds and a binary representation outside of the inter-threshold interval, or fitting parametric S-shaped functions with nonlinear least squares. All these methods lack flexibility and capability of extrapolation, and are better suited for studies over small areas. The simple binary description is effective in its simplicity when the researcher is interested in particular in extreme events or long run total climatologies, but it cannot provide a reliable reproduction of the entire snowfall distribution. Moreover, the thresholds in the first two methods are often established simply by looking at the plot of snow fraction against temperature, or from statistical analysis of the entire available dataset all at once. However, in case of gridded data over large areas, the optimal threshold values may vary sensibly depending on the location.

We applied a breakpoint search algorithm to find point-specific values of the threshold temperature(s), finding distinct areas where the optimal number of thresholds is 1 or 2. We then used the results to inform different regression models: a segmented linear regression on the logit-transformed snow fraction, a spline regression on the logit-transformed snow fraction, and a piecewise beta regression with a logit link. The latter resulted not enough numerically stable, so that the model could be applied only to a limited number of grid points, and hence discarded. The segmented and spline regressions, on the other hand, produced consistent and very similar results over the entire domain.

We used ERA5 reanalysis over Europe for the period 1979-2018 for model selection, by estimating each model at each grid point over a training set, and comparing the performances in terms of prediction of out-of-sample observations. Compared to regressions without thresholds and to the naive binary classification, both models performed markedly better in terms of mean absolute error, root mean square error and Pearson's correlation between observation and prediction.

We then considered the historical period 1979-2005 of the IPSL_WRF high resolution climate simulation model, for which bias corrected near surface temperature and total precipitation are available, while snowfall is un-adjusted, showing very large biases with respect to ERA5, especially over areas characterized by frequent and abundant snowfall. We find that the point-specific model estimates obtained in the model selection phase can be applied to this dataset to obtain snowfall estimates much closer to reanalysis compared to the raw IPSL_WRF data. In particular, we show that the reconstructed snowfall improves remarkably in terms of total long run statistics and divergence between probability distributions with both methods, proving that these can be used in place of more complex multivariate bias correction schemes. We provide detailed results for the Alps region and Norway, two areas characterized by very large total snowfall and extreme biases at the same time. We show that, over these regions, the bias in total snowfall is reduced by one order of magnitude, while the proximity between model and reanalysis distributions improves dramatically.

Overall, there are no clear indications leading to the choice of either model between segmented and cubic spline regression in terms of capability to reconstruct the snowfall. We argue that cubic spline regression may be a better choice in practice because it is more flexible in describing nonlinear relationships, and it can be better extrapolated to values that are not in the observed training data. Moreover, we recall that segmented regression performs a re-estimation of the threshold temperatures, but it is still required to run the breakpoint search algorithm, so that meaningful starting points can be provided. On the other hand, spline



regression shows a satisfactory performance directly using the breakpoint analysis results. As a final remark, we underline that while we chose to use the one or two threshold temperatures as knots for spline regression, very different approaches can be used to assign knot values, for example using deciles of the distribution of the independent variable. We decided to limit our analysis to the case of knots corresponding to up to two threshold temperatures for coherence with the previous literature and better interpretability, but more flexible approaches may be used in cases where it appears to be necessary.

## 6   Acknowledgments

The authors thank Saverio Ranciati for useful discussion. This work is supported by the ANR-TERC grant BOREAS.

*Competing interests.*   The author declares no competing interest.



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




**Figure 1.** Left column: 1979-2005 total DJF snowfall (m), right column: 1979-2005 number of DJF snowfall days. Upper panels: ERA5, lower panels: IPSL_WRF.

**Figure 2.** Results of the breakpoint analysis. Upper panel: locations with a single threshold temperature (a). Lower panels: locations with double threshold temperatures; lower thresholds (b), upper thresholds (c).


**Figure 3.** Threshold temperatures optimized for segmented regression according to Eq. (11). Upper panel: locations with a single threshold temperature (a). Lower panels: locations with double threshold temperatures; lower thresholds (b), upper thresholds (c).





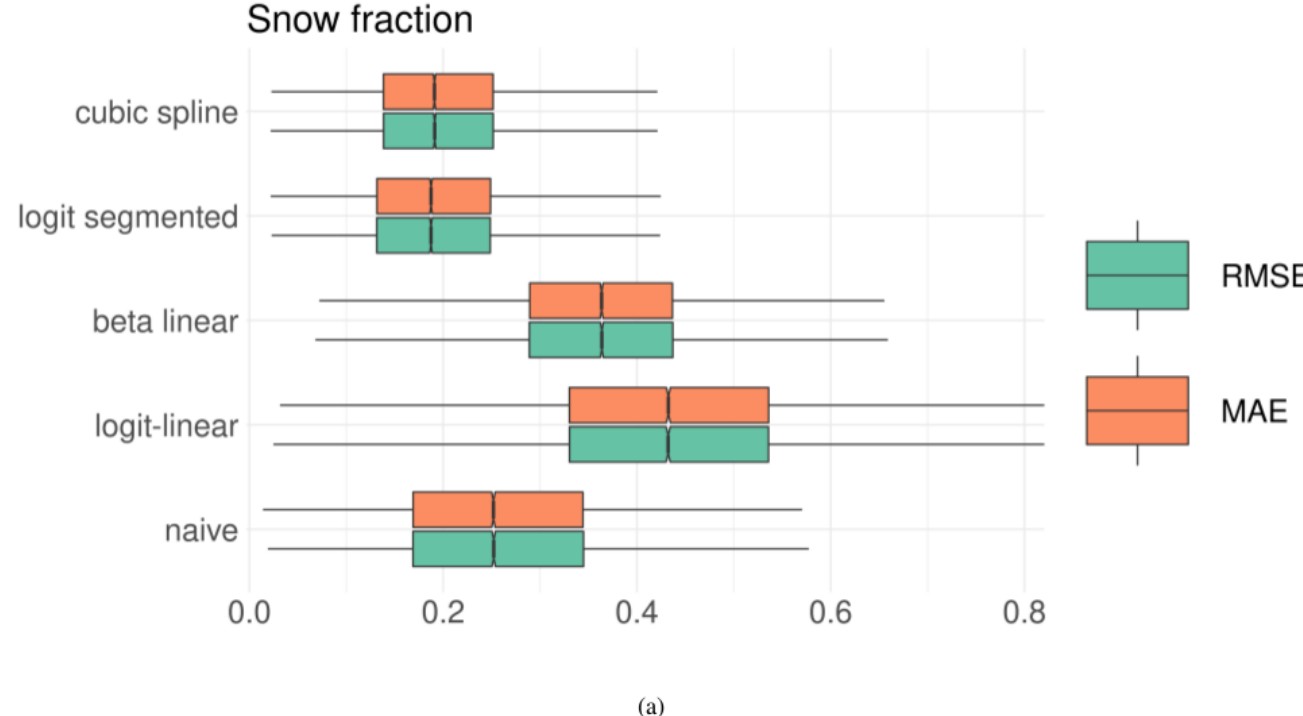

(a)

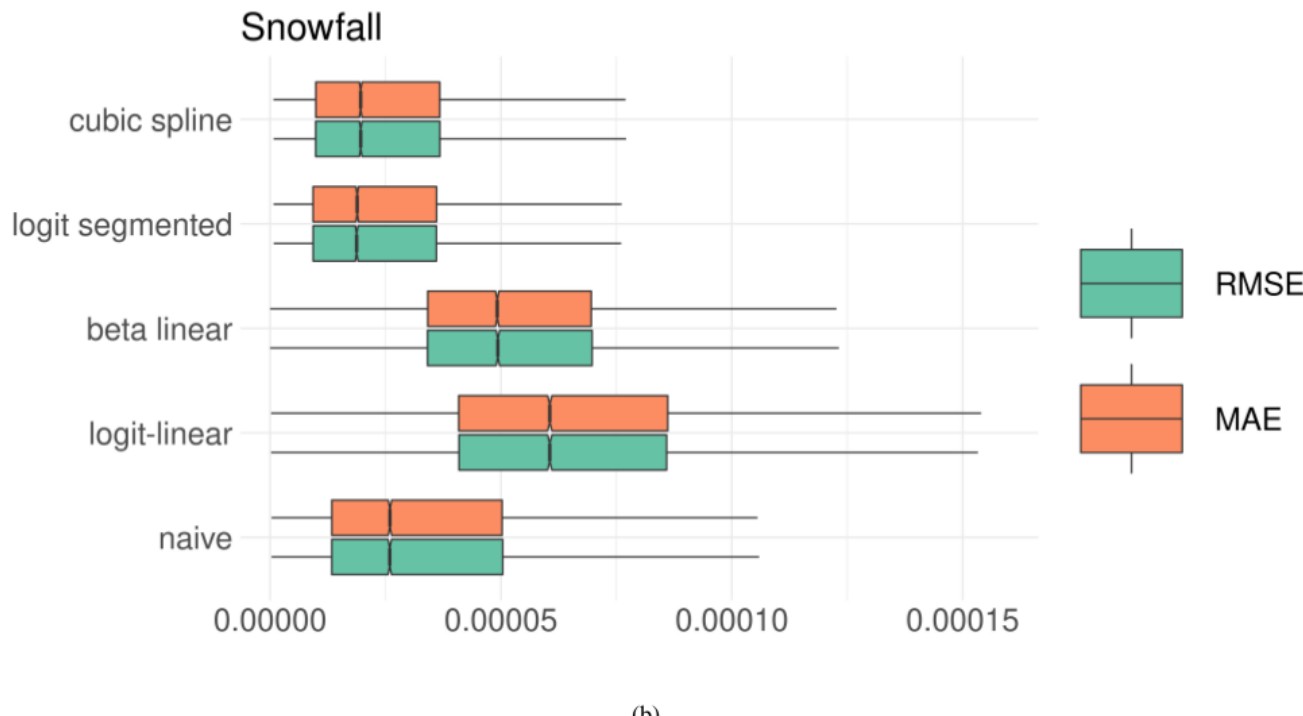

(b)

**Figure 4.** Forecasting performance of the 5 considered models in terms of mean absolute error (orange) and root mean squared error (green). Upper panel: snow fraction, lower panel: snowfall.

**Figure 5.** Boxplots of correlation coefficients between the snowfall reanalysis and predicted using the 5 selected models, for all grid points with at least 30 snowfall events (a). Mapped values of the snowfall correlation coefficient for the segmented regression (b), cubic spline regression (c), naive threshold model (d).

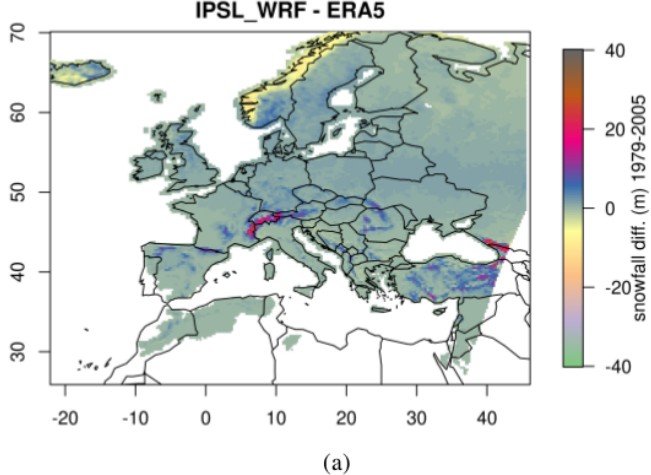

(a)

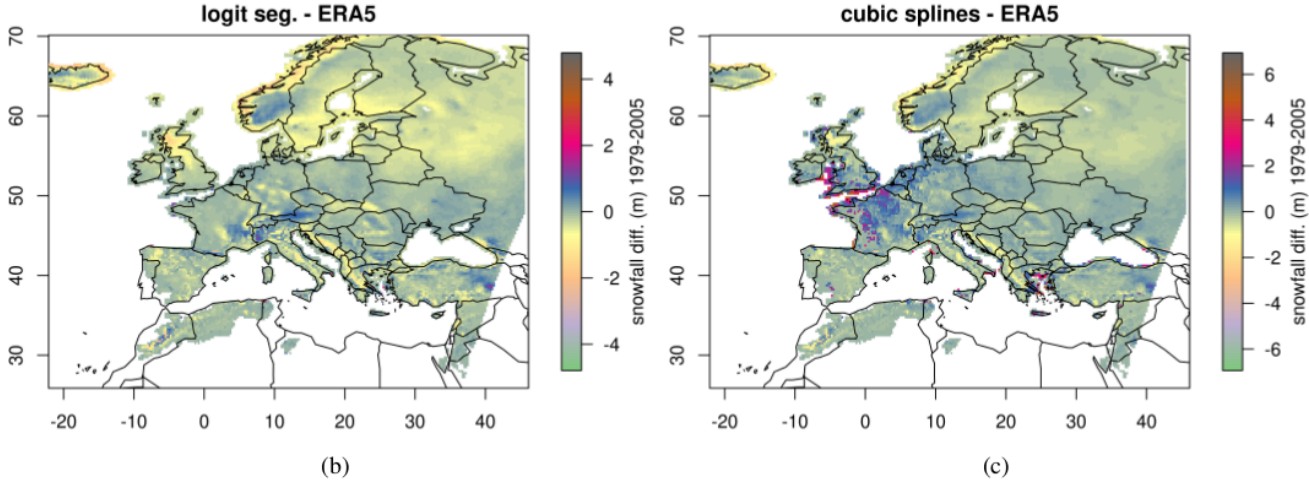

(b)                                              (c)

**Figure 6.** Difference in meters between model and reanalysis total snowfall for the years 1979-2005; IPSL_WRF model (a), segmented regression (b) and spline regression (c).



**Figure 7.** Performance of the statistical models as bias correction methods in terms of statistical divergences. Boxplots of the three chosen divergence measures for the statistical models and IPSL_WRF raw simulations (a); maps of the KS statistic differences between the two model and raw IPSL_WRF (b); comparison of the KS probability distributions in the three models.

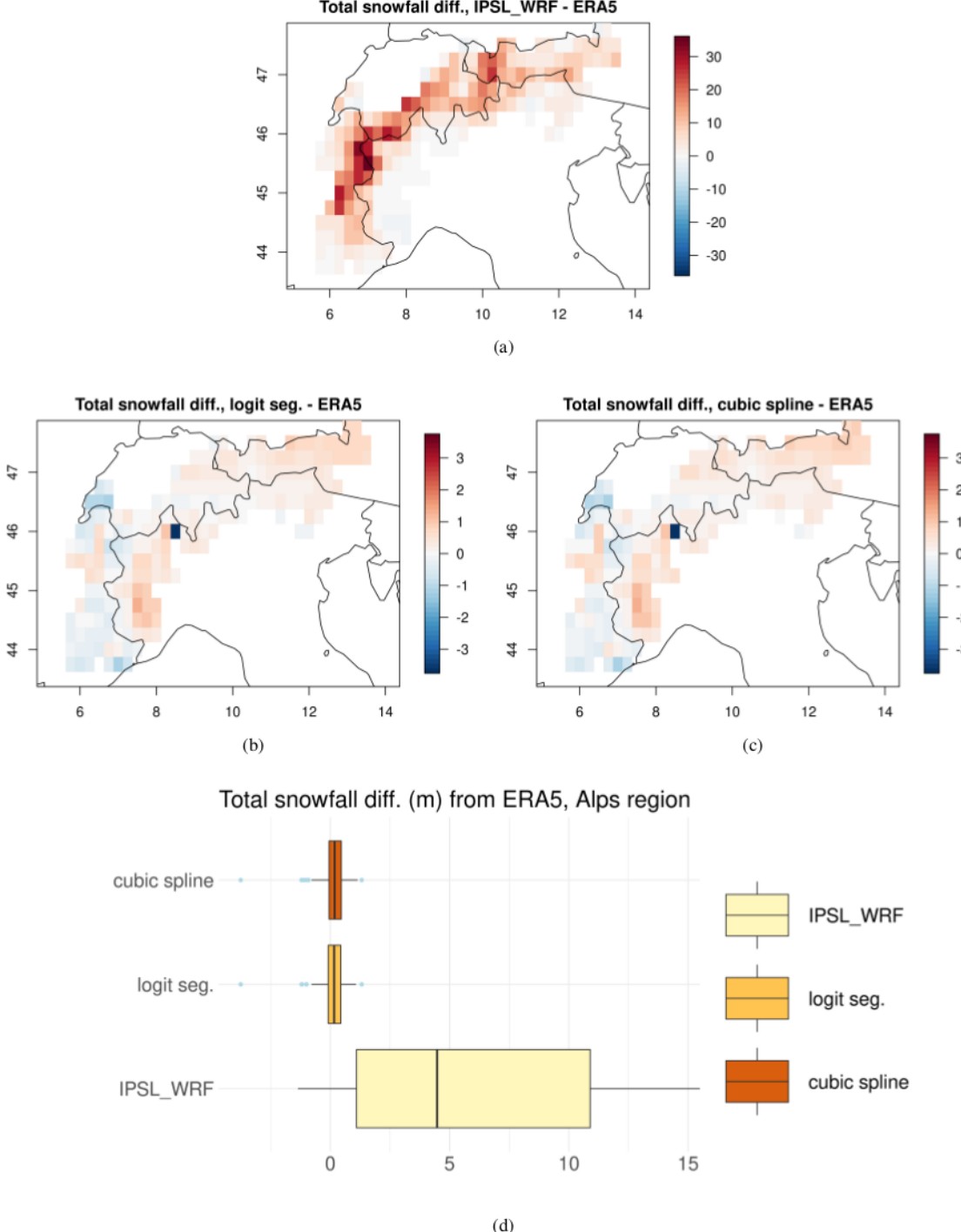

**Figure 8.** Maps of total 1979-2005 snow differences over the Alps region between IPSL_WRF (a), segmented regression (b), cubic spline regression (c) respect to ERA5. Corresponding summary boxplots are shown in panel (d).


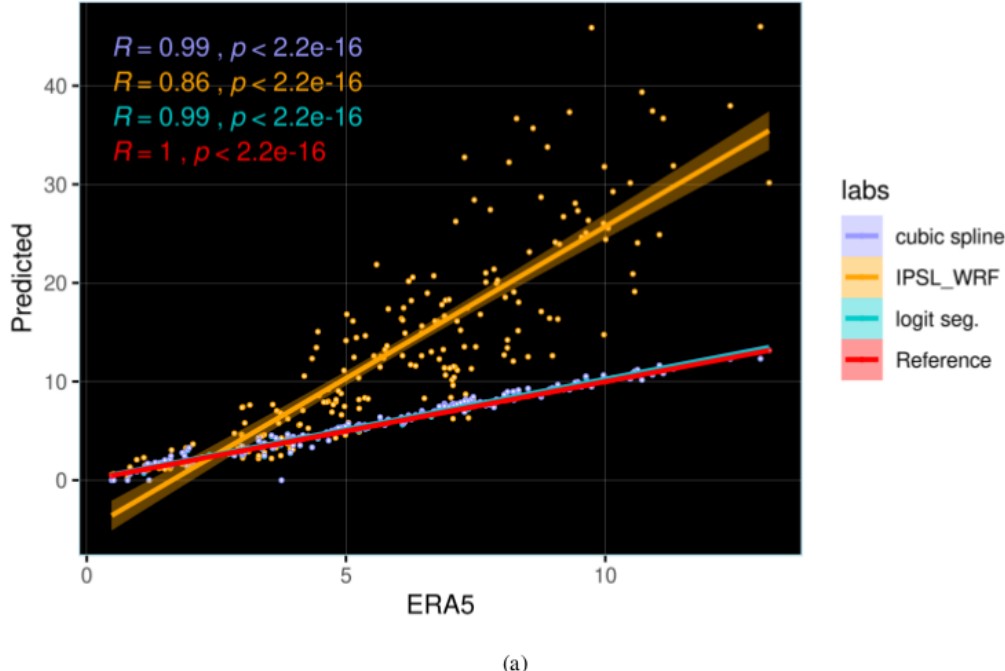

(a)

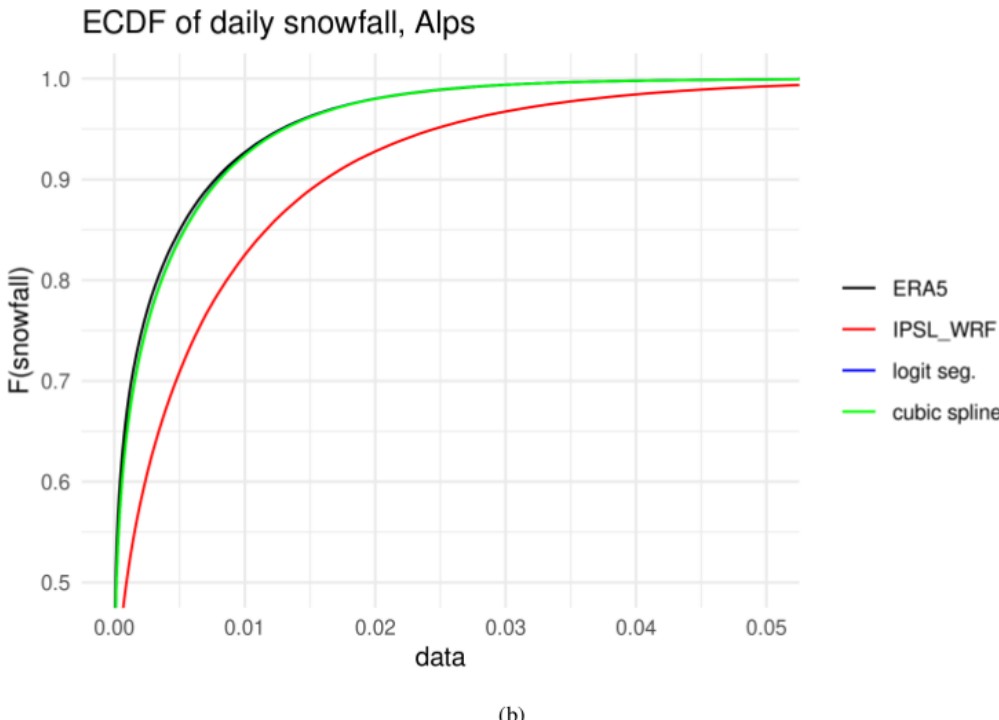

(b)

**Figure 9.** Upper panel: correlation between total 1979-2005 snowfall over the Alps region in ERA5 and IPSL_WRF, segmented regression, cubic spline regression results. Lower panel: comparison between the empirical distribution functions of reanalysis and modelled daily snowfall over the Alps region.





**Figure 10.** Maps of total 1979-2005 snow differences over the Norway between IPSL_WRF (a), segmented regression (b), cubic spline regression (c) respect to ERA5. Corresponding summary boxplots are shown in panel (d).




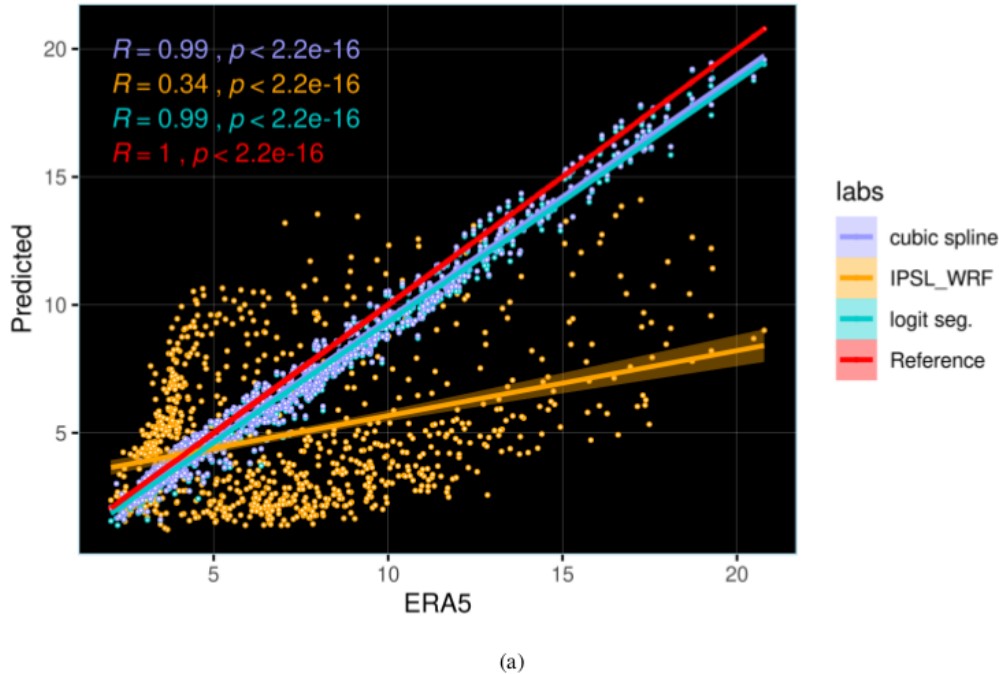

(a)

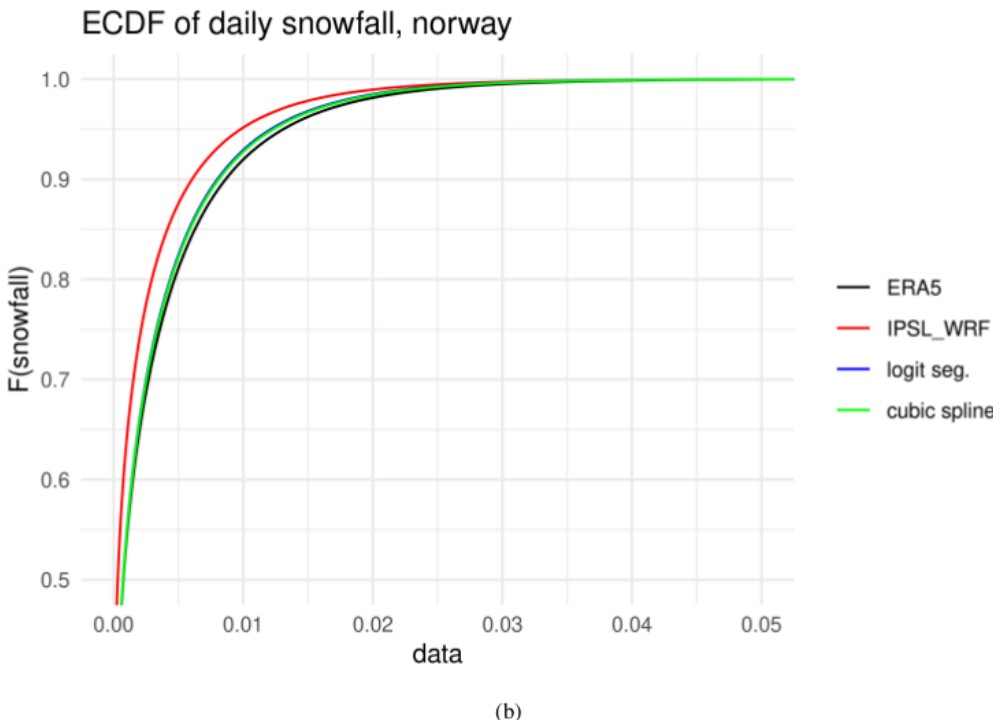

(b)

**Figure 11.** Upper panel: correlation between total 1979-2005 snowfall over Norway in ERA5 and IPSL_WRF, segmented regression, cubic spline regression results. Lower panel: comparison between the empirical distribution functions of reanalysis and modelled daily snowfall over Norway.