# Peer review of "Improving snowfall representation in climate simulations via statistical models informed by air temperature and total precipitation"

_Natural Hazards and Earth System Sciences, 2020_

## Referee Comment (RC1) · Anonymous Referee #1 · 27 Dec 2020

The manuscript by Pons and Farada assesses the performance of several snowfall separation methods to reproduce simulated snowfall in the ERA5 reanalysis on a European scale by taking into account simulated near-surface air temperature and total precipitation at daily resolution. The two best-performing methods are in a second stage applied to bias-adjusted output of the IPSL-WRF regional climate model (historical period) to obtain a bias-adjusted estimate of simulated snowfall in the RCM. The evaluation reveals a satisfying representation of the PDF of the daily ERA5 reference snowfall amount in the historical period by the bias-adjusted and separated IPSL-WRF

simulation.

Overall, the paper fits well into the journal's scope. Data and methods are for most parts clearly introduced and explained. The presentation of the results has some weaknesses but is still acceptable. The major drawback of the work, however, is the unclear relevance of the work for a broader audience and for RCM snowfall bias-adjustment. Essentially, the authors search for a method to emulate the ERA5 microphysics scheme that simulates the actual snowfall flux in the reanalysis model taking into account simulated near-surface temperature and simulated total precipitation only. The two best performing methods are then applied to a different model (IPSL-WRF) to separate snowfall from total precipitation after bias-adjustment of simulated temperature and precipitation. Results look satisfying, but there is

(1) no evaluation of the ERA5 snowfall flux (which is the basic reference in the entire work, and the entire analysis is geared towards a reproduction of ERA5-simulated snowfall flux; the paper frequently uses the term "observed" for ERA5 snowfall flux, although it is essentially a simulated flux probably subject to systematic biases)

(2) no analysis to what extent the satisfying results of the application of the method to the RCM are specific for the chosen RCM and the bias-adjustment method of temperature and precipitation that was carried out beforehand (a different RCM might, even after bias-adjustment, have a completely different multi-variate structure of daily temperature and precipitation, at least a structure that is different to ERA5, and the method might not hold in these cases

(3) no discussion of potential problems with inter-variable dependencies even after bias-adjustment of an RCM (-> see, for instance, Meyer et al., HESS, 2019)

(4) no indication if the identified methods will also produce robust snowfall estimates in a future climate change scenario (which is, as far as I can guess, the basic motivation of the entire work -> a possibility to investigate such an applicability would be to split the ERA5 period into "cold" and "warm" years and to calibrate on the cold and validate

on the warm sample)

(5) no reference to differences in spatial resolution of the models employed and the fact the subgrid orography can actually have a considerable influence on simulated snowfall (or, the other way round, neglecting subgrid scale orographic variability in model bias-adjustment could results in false derived snowfall sums)

(6) no analysis of a calibrated threshold within the "naive" STM method (which I assume could yield even better results than the two best identified methods, as even the performance with a fixed 2°C threshold is very close to the two best-performing methods), and

(7) no analysis of the importance of variations on the sub-daily scale which might be important for daily snowfall sums.

The main message of the manuscript is currently, that for this specific setup (this specific RCM, this specific bias-adjustment method, this specific reference snowfall), the two identified methods if applied to bias-adjusted IPSL-WRF temperature and precipitation output can yield a representation of snowfall that well reproduces the ERA5 reference snowfall. These results are in my opinion not per se transferable to different models or to a future climate scenario or to a different reference snowfall (especially not to a true observation-based snowfall estimate). As such, the value of the work is limited for the time being in my opinion and not too informative for a broader readership. I would hence recommend to return the manuscript to the authors for major revisions. During these revisions, the mentioned points should be picked up in order to increase the relevance of the work. A couple of further issues are mentioned below.

With kind regards.

FURTHER ISSUES:

Line 24: Very unclear what is meant.

Lines 35-36. Also rather unclear.

Lines 38-40: This is actually not true, the entire set of so-called "perfect progno-sis" downscaling methods is ignored here. These do not adjust the simulated vari-ables towards observations but exploit calibrated relationships between observed (or reanalysis-simulated) large scales and observed local scales.

Lines 141-142: Very unclear.

Line 145: Above (line 132) you mention that only daily data are used, here you obvi-ously employ hourly data. Please clarify.

Line 151: "grid step" unclear

Line 181: Rather unclear what is meant by "standardized temperature anomalies" and why these are used.

Chapter 3.1.2: This sub-chapter contains a large amount of rather technical informa-tion, which is appreciated, but which should be moved to some technical appendix I believe.

Line 517: Do you have any explanation for these rather low calibrated tresholds? Is there a relation to orographic height, for instance?

Line 526: Should be "Fig. 2" instead of "Fig. 1".

Lines 688-689: Very unclear.

Lines 707-708: Better representation of the tails is not really apparent from the figure I'd say.

Figure 1: Color scale is not very intuitive.

Figures 2 and 3: Bad color scale: White color means threshold temperatures around 0°C but also "not applicable". I'd suggest to modify the color scale.

Figure 4: Legend too large. Also, the methods are named differently compared to Table 1 and are sorted in a different order. Please harmonize. Also, it would be good to use

the same unit in the lower panel as in Table 1 (10ˆ-3)

Figure 5: Upper panel: Please use the same sorting of methods as in Table 1.

Figure 6: Very bad color scale, not at all intuitive. Also, the color scale should be identical for all panels to enable a comparison (same color should mean the same value in all panels). Is the unit actually m/27 years (1979-2005) or m/year? Please clarify.

Figure 7: Legend of lower panel too small.

Figure 8: What about the bad-performing grid cell in northern Italy in logit seq and cubic spline? What is happening here?

Figures 9 and 11, upper panels: Sorry, but even after reading the explanation several times it is not really clear to me what is displayed here. Also, I'd suggest to use a white background instead of a black background. Lower panels: Please specify the unit of the x-axis.

---

## Referee Comment (RC2) · Anonymous Referee #2 · 30 Jan 2021

Ponds and Faranda present and assessment of statistical models in order to better represent the snowfall using data from gridded-climate products. The assessments include Europe and comprise the winter months of December, January and February. ERA5 reanalyses between 1978 and 2018 and IPSL-WRF between 1979 and 2005 were used to evaluate the statistical models. Overall, the methods presented improve the representation of the snowfall, and due to the characteristics of the methods, an important advantage is its applicability to larger areas using gridded climate products. Although the overall research fit in the journal, there are several points that need to be

addressed. More importantly, I think that a better rationale for this works is necessary as well as some changes in the structure and/or a reduction in the length. In view of this, I suggest mayor revisions. I do really hope that this review will be useful to the authors to improve their manuscript.

It is not clear why is important to use these statistical models. Extreme events and its impacts (e.g. February 2018) are barely mentioned in the Introduction and also the limitations in represent snowfall events of the Climate models using emission scenarios. What is the main goal of applying these methods? For the experiment presented, it seems that is to find a better representation in climate change models scenarios. But, It is applicable to weather forecast as GFS for instance? It would be useful to have a better idea of why this work is important. To extend the rationale of the work presented.

Some sections are hard to read. The Introduction seems more appropriate for a review paper, maybe avoid the details of each earlier model. Method section also seems lengthy and hard to follow, I suggest to try to go to the point. Results also contain information that is not appropriate for this section (see below). I suggest re-restructuring the manuscript, and move some sections to a Supplementary Material in order to reduce the length of the manuscript. Maybe a Discussion section will do the manuscript more clearer moving some Results and Conclusion paragraphs to this section, allowing concentrate (and reducing) the Conclusions to the main findings.

Additionally, please note that some literature refers to the "separation of snowfall" as "precipitation‐phase partitioning methods (PPMs)" (e.g. Harder and Pomeroy, 2014). Consider in to use this terminology.

Other comments (line number indicated):

16: This was already mentioned in Line 12.

25-30: Add a more specific statement about why statistical models are better than physics parameterization or simplifications in climate models. Is this just a thought? or

Do you have more evidence of this? some references?

50: "Hemispheric".

58-59: Worth to mention that solid precipitation also depends on relative humidity and could be useful to estimate the mixed-phase or sleet (e.g. Ding et al., 2014).

116: Change "Section ??" to "Section 4".

125-128: This paragraph sounds like a better justification of the work and fits with the overall aim of the journal. Consider to move to the Introduction section and explain a bit more the justification of this study.

134-135: The election of these months is a bit contradictory if the aim is to analyse extreme events causing disruption. Of course, extreme events largely occur in winter, but over other seasons (end of Autumn, the beginning of Spring) extreme snowfall events could occur. For instance, the so-called "Beast from the East" in 2018 was between the end of February and beginning of March. If analysis months is not extended maybe change the title to "Improving winter snowfall representation. . ."

147: Delete extra "( )".

241: Numbering missing.

246: Delete extra "( )".

509: Fig. 2 is mentioned here before Fig. 1. Also change to "Fig. 2" to be consistent with the rest of the manuscript.

526: You refer here to Fig. 2 I guess not Fig. 1.

551: Most of section 4.2 seems more appropriate to the Methods section (or Supplementary Material). For instance, lines 575-585 is an explanation of the Kruskal-Wallis H test, lines 593-601 is an explanation of the U-test. Here, you must show the results after applying these tests.

536: Numbering missing.

737-755: These paragraphs are a repetition of previous statements already presented in the Introduction and related to limitations of previous methods, observational data and the physics of the climate models. I think is not necessary to repeat in the Conclusions.

Figure 7d): Legend is not visible.

Figures: please add units and names to the axis where these are missing.

References

Ding, B., Yang, K., Qin, J.,Wang, L., Chen, Y., and He, X. (2014). The dependence of precipitation types on surface elevation and meteorological conditions and its parameterization. J. Hydrol. 513,154–163. doi: 10.1016/j.jhydrol.2014.03.038

P. Harder, J.W. Pomeroy (2014) Hydrological model uncertainty due to precipitation-phase partitioning methods Hydrol. Process., 28 (2014), pp. 4311-4327, 10.1002/hyp.10214

---

## Referee Comment (RC3) · Anonymous Referee #3 · 1 Feb 2021

Review:

Improving snowfall representation in climate simulations via statistical models informed by air temperature and total precipitation (2020)

By Pons and Faranda

General Comments:

In this manuscript, the authors test existing methods for the estimation of the apportionments of rain/mixed/snowfall and present a more robust way of selecting threshold

temperatures to be used in single/multiple threshold models. The tests are performed based on the reanalysis ERA5 dataset and the IPSL_WRF climate projection model for the period 1979-2005, and at 0.25-degree spatial resolution in Europe, including the Scandinavian peninsula. Daily temperatures are used for the models.

In general, I find that the authors did a good job explaining the different methods and performance analyses, and as a data exercise, the procedures are somehow clear in the manuscript. However, I find that there is a disconnect between the objectives and how realistic it would be to apply these models over such domains and spatial resolution. I also find that the use of the reanalysis data as an "observation" (e.g., Figure 5) to test all the models against can easily be challenged given the uncertainties in such datasets, especially to determine snowfall in complex terrain.

I would like to point out that the type of models that the authors are applying are generally derived from meteorological data at in-situ stations, while the article refers to modeling grid scales of several km (0.25 degrees, which at 70 deg lat is $\sim$ 10 km in E and 28 km in N coordinates, https://www.opendem.info/arc2meters.html). How can the same type of models be appropriate for both scales? A gridcell of such dimensions would easily cover a mountain valley and surrounding mountains, with a very wide elevational range. Throughout my reading, I just kept wondering how such models could be applied in this spatial context. One would expect that over the domain of a single gridcell, one would potentially encounter a portion of the area to receive rainfall, while another portion would receive snowfall and a middle portion would be in the transitional zone if an event occurs around the freezing point. I find this to be very problematic in the context of the manuscript.

Another relevant issue has to do with the use of daily temperatures for the models, given that sub-daily temperature fluctuations would have marked effects on precipitation phase apportionments. Also, what would the near surface temperature be representative of in the manuscript? A mean elevation? If so, would this be realistic? I argue it is not, especially to illustrate how the proposed methods can enhance the estimation

of the model's parameters.

As snow hydrologists, in our group's modeling efforts we use a similar two-temperature threshold model to estimate the precipitation apportionments at grid scales between 10-100 m in mountainous terrain, with a linear model between the two thresholds. Even at such scales, we understand that there are drawbacks to such model, but the uncertainties in precipitation amounts and temperatures are primary, and the model for the precipitation apportionments takes a secondary role. However, even at such scales, determining the performance of the model and parameters is very challenging because of the difficulties in determining accurate precipitation amounts, particularly snowfall.

Threshold values also seem unrealistic in some locations, as low as -15 or as high as 5+ C. There are examples of snowfall events at high temperatures (e.g., late June 2019 summer events in Colorado), but as a generalized modeling threshold it would seem unrealistic to have such high and/or low values. This would also highlight the issues regarding the data and spatial scales of the analysis.

Because of these issues, I am recommending that the manuscript be rejected. I ultimately consider that the results in the manuscript do not accomplish demonstrating how the proposed methods deliver improvements in model accuracy.

Specific Comments:

ll. 35-36: Odd sentence.

l. 52: typo in "observationa".

ll. 74 & 79: I suggest using past tense to refer to findings or proposed models, as the ones in these lines (Pipes and Quick (1977) and L'hôte et al. (2005)).

l. 116: typo "Section ??"

l. 122: I suggest adding a comma before "which", but the sentence would need

changes.

l. 125: what are you calling "large scales" here? Suggest clarifying.

l. 141: "worth mentioning".

l. 291: suggest changing "It is easy to prove" to "It can be shown".

l. 259: Revise "The latter resulted not enough numerically stable".

---

## Author Comment (AC1) · 14 Mar 2021

General Comments: In this manuscript, the authors test existing methods for the estimation of the apportionments of rain/mixed/snowfall and present a more robust way of selecting threshold temperatures to be used in single/multiple threshold models. The tests are performed based on the reanalysis ERA5 dataset and the IPSL_WRF climate projection model for the period 1979-2005, and at 0.25-degree spatial resolution in Europe, including the Scandinavian peninsula. Daily temperatures are used for the

models. In general, I find that the authors did a good job explaining the different methods and performance analyses, and as a data exercise, the procedures are somehow clear in the manuscript. However, I find that there is a disconnect between the objectives and how realistic it would be to apply these models over such domains and spatial resolution.

A: We are aware that these statistical models have been initially formulated for, and applied to, the estimation of snowfall from station data, and that it would be ideal to work at very small scales to catch all modes of spatio-temporal variability of any given phenomenon (not only in atmospheric sciences).

This is true at all levels, in fact any simulation model for meteorology, climate, oceanography, land-atmosphere interaction and so on is integrated at a finite time step that necessarily cuts off high temporal frequencies, and over finite grids that require sometimes coarse approximations of the sub-grid scale processes. It is not ideal, but if scientists in the climatology field chose to wait until it will be possible to integrate climate models at the molecular dissipation scales, present climate studies would not be able to address much more than global averages. As climatologists, we must deal with these approximations, and do our best to mitigate their effects.

Q: I also find that the use of the reanalysis data as an "observation" (e.g.,Figure 5) to test all the models against can easily be challenged given the uncertainties in such datasets, especially to determine snowfall in complex terrain. I would like to point out that the type of models that the authors are applying are generally derived from meteorological data at in-situ stations, while the article refers to modeling grid scales of several km (0.25 degrees, which at 70 deg lat isâĹij10 km inE and 28 km in N coordinates, https://www.opendem.info/arc2meters.html).

A: We agree that the term "observation" is incorrect in this context, and will be replaced by "reanalysis" or "reference dataset" in the future version of the paper.

Q: How can the same type of models be appropriate for both scales? A gridcell of such

dimensions would easily cover a mountain valley and surrounding mountains, with a very wide elevational range. Throughout my reading, I just kept wondering how such models could be applied in this spatial context. One would expect that over the domain of a single gridcell, one would potentially encounter a portion of the area to receive rainfall, while another portion would receive snowfall and a middle portion would be in the transitional zone if an event occurs around the freezing point. I find this to be very problematic in the context of the manuscript. Another relevant issue has to do with the use of daily temperatures for the models,given that sub-daily temperature fluctuations would have marked effects on precipitation phase apportionments.

A: The reviewer has concerns about the idea that these models are applied to gridded datasets with a relatively coarse resolution and daily frequency. However, as also mentioned in the literature review in Section 1, this type of model is already applied in the recent climate literature focusing on snowfall. For example, Bai et al., 2019 consider observations projected on a $0.25°$ grid, while Chen et al., 2020 use a sigmoid function to estimate snowfall data from GCM simulations, at a $1.5°$ resolution. A comparison of the performance of the simple single threshold method to estimate snowfall over Europe, compared to E-OBS, can be found in Faranda 2020. All of these studies consider daily data, as well as others dealing with station data, such as Liu et al., 2018.

One may rather wonder if, given the level of approximation present in climate simulations, the use of S-shaped functions is inappropriate, and simpler models such as the binary threshold should not be applied instead. This is indeed the research question we addressed in the paper: the results in terms of reconstruction of the snow in the reference period suggest that models admitting a nonlinear function provides better performances than naive binary apportionment or simple linear regressions, and we think that this point is clearly proven by the results discussed in Section 4 of our paper.

This happens, despite all the concerns about neglected complexity due to sub-grid scales, because the relationship between snow fraction and temperature can be seen

as a transition between two fixed boundaries (i.e. a snow fraction equal to 0 and 1), regardless of the coarse spatio-temporal resolution. Most smooth transitions of this type result in S-shaped relationships, regardless of the scales involved. For example, analogous S-shaped transitions can be observed when studying turbulence in a nocturnal stable boundary layer (Van de Wiel et al., 2017), involving different processes and much smaller scales than in snow hydrology.

We do not find this fact particularly surprising. Statistical models are always wrong with respect to the reality of the phenomenon they try to catch: their efficacy must be measured by how useful they are in terms of the goal, in this case the reconstruction of an unobserved variable from two observed covariates (technically, a prediction task). Since our goal is purely predictive, the effectiveness of the method should be judged based on their prediction performance, and not comparing it to the way it is used in different contexts.

The fact that the same statistical model can be used at very different scales (or even in completely different fields, or branches of a field) is not an exception in quantitative research. A time series model featuring intermittency can catch features of the generating process in small-scale turbulence as well as at the climate scale; econometric models can be applied to study the micro-performance of a single agent as well as of entire countries; exponential or gamma laws describe the waiting times between two quantistic as well as macroscopic events; Lotka-Volterra equations can be used to model the relationships both between populations of large prey-predators, and between small pathogens such as phages and bacteria, et cetera.

The nature of the phenomenon suggests that an S-shaped relationship could perform better than anything less smooth such as a binary threshold, and we believe that our results show that this is indeed a realistic conclusion.

Q: Also, what would the near surface temperature be representative of in the manuscript? A mean elevation? If so, would this be realistic? I argue it is not, especially to illustrate how the proposed methods can enhance the estimation of the model's parameters.

A: As standard in gridded climate simulations and reanalysis, the near-surface temperature is representative for the mean elevation in the grid cell. Again, we understand how these scales may seem disproportionately large when compared to the domain of a single weather station, but this logic would basically exclude using climate simulation models for anything but computing yearly/global statistics.

Q: As snow hydrologists, in our group's modeling efforts we use a similar two-temperature threshold model to estimate the precipitation apportionments at grid scales between 10-100 m in mountainous terrain, with a linear model between the two thresholds. Even at such scales, we understand that there are drawbacks to such model, but the uncertainties in precipitation amounts and temperatures are primary, and the model for the precipitation apportionments takes a secondary role. However, even at such scales, determining the performance of the model and parameters is very challenging because of the difficulties in determining accurate precipitation amounts, particularly snowfall.

A: We find that this consideration indeed marks one of the greatest differences between a climate study and a small scale study. Of course there is a lot of uncertainty also in the performance of reanalysis models, in addition to the coarse graining of the process. However, when conducting an analysis on climate simulation model outputs, the reference period is chosen to be a previously validated gridded observation or reanalysis dataset. Given the validation, the dataset is taken as the reality, and used to bias correct the climate models. It is even possible to conduct a so called "perfect model" experiment, where the output of a climate model is used as the reality to check the performance of the bias correction on another model.

While we remain aware of the limitations of reanalysis datasets, it is once again a level of approximation that we must accept, as the alternative would be to drop climate

studies altogether.

In such a framework, after validation against another reliable dataset, we neglect uncertainty in the reanalysis and we use it for the bias correction of the climate models. At this point, we assume that the bias corrected variables in the model are unbiased. This removes the issues of observational error in each model, and the use of model ensembles (at least tries to) mitigate the effects of the model-specific approximations.

Clearly, the resulting outputs cannot be considered representative of sub-grid scales. However, we do consider the bias corrected variables representative for the resolved scales. The issue with snowfall, as stressed in the paper, is that it poses more challenges than other variables in terms of direct bias correction, while the raw data are not realistic, and they also become incompatible with temperature and total precipitation once these two variables are bias corrected. Our objective is not to use models that represent correctly subgrid scales, but simply to leverage on variables that are relatively easy to adjust and provide a better reconstruction of the snowfall compared to raw data.

We remark again that our goal is purely predictive and the effectiveness of the method should be judged based on results in these terms, and not comparing it to the way it is used in different contexts. Not only results discussed in Section 4.2 - 4.3 clearly point to an improvement of the representation of snowfall in these terms, but such improvement is particularly dramatic over areas characterized by complex orography, as shown by the case studies focusing on the Alps and Norway, showing that indeed these models work well even at a coarser level.

Q: Threshold values also seem unrealistic in some locations, as low as -15 or as high as 5+ C. There are examples of snowfall events at high temperatures (e.g., late June 2019 summer events in Colorado), but as a generalized modeling threshold it would seem unrealistic to have such high and/or low values.

A: We agree, in many areas the estimated thresholds are not directly interpretable in

meteorological terms. However, our objective is not to obtain results that reproduce realistic microphysics, but to estimate models that provide the best predictive power. The choice of the knots for spline regression is completely arbitrary: for example, one common choice is to use the deciles of the covariate distribution. We choose to use breakpoint analysis with up to 2 thresholds because we expect relatively smooth and monotonic relationships, but we do not imply that the recovered thresholds are necessarily realistic for every grid point.

We also stress that we presented results from two different techniques, showing that actually Eq. 11 provides thresholds that are more physically realistic, while the unadjusted breakpoint analysis provides less interpretable values. Please also notice that, since we work with standardized anomalies (due to the different scales of the variables involved in the regressions) as specified in the article, these values are not meant to be interpreted as absolute temperature, but as deviations from the long-term climatology.

Q: This would also highlight the issues regarding the data and spatial scales of the analysis. Because of these issues, I am recommending that the manuscript be rejected. I ultimately consider that the results in the manuscript do not accomplish demonstrating how the proposed methods deliver improvements in model accuracy.

A: We are frankly puzzled by this decision, and especially by the way it is justified. The only comment actually referring to the results concerns the threshold temperatures, an instrumental value obtained at the beginning of the modelling procedure, and not one of the primary objectives of our analysis. This last comment leaves us with the impression that the reviewer formed their opinion based on their previous knowledge about these models (applied in a different context) and did not take into proper account the results that should indicate whether or not we answered the initial research question. We underline once again that we propose an improvement of techniques that are already used for similar or analogous tasks, so rejecting the present paper on this basis means to also challenge part of the existing literature.

Q: Specific Comments:

A: Thank you, we will make sure to make all the corrections listed below.

ll. 35-36: Odd sentence.l.

52: typo in "observationa".

ll. 74 & 79: I suggest using past tense to refer to findings or proposed models, as theones in these lines (Pipes and Quick (1977) and L'hôte et al. (2005)).

l. 116: typo "Section ??"

l. 122: I suggest adding a comma before "which", but the sentence would need changes.

l. 125: what are you calling "large scales" here? Suggest clarifying.

l. 141: "worth mentioning".l.

291: suggest changing "It is easy to prove" to "It can be shown".

l. 259: Revise "The latter resulted not enough numerically stable".

References

Bai, L., Shi, C., Shi, Q., Li, L., Wu, J., Yang, Y., ... & Meng, J. (2019). Change in the spatiotemporal pattern of snowfall during the cold season under climate change in a snow‐dominated region of China. International Journal of Climatology, 39(15), 5702-5719.

Chen, H., Sun, J., & Lin, W. (2020). Anthropogenic influence would increase intense snowfall events over parts of the Northern Hemisphere in the future. Environmental Research Letters, 15(11), 114022

Liu, S., Yan, D., Qin, T., Weng, B., Lu, Y., Dong, G., & Gong, B. (2018). Precipitation phase separation schemes in the Naqu River basin, eastern Tibetan plateau. Theoretical and applied climatology, 131(1), 399-411.

Van de Wiel, B. J., Vignon, E., Baas, P., van Hooijdonk, I. G., van der Linden, S. J., Antoon van Hooft, J., ... & Genthon, C. (2017). Regime transitions in near-surface temperature inversions: A conceptual model. Journal of the Atmospheric Sciences, 74(4), 1057-1073.

---

## Author Comment (AC2) · 14 Mar 2021

Pons and Faranda present and assessment of statistical models in order to better represent the snowfall using data from gridded-climate products. The assessments include Europe and comprise the winter months of December, January and February. ERA5 reanalyses between 1978 and 2018 and IPSL-WRF between 1979 and 2005were used to evaluate the statistical models. Overall, the methods presented improve the representation of the snowfall, and due to the characteristics of the methods,

an important advantage is its applicability to larger areas using gridded climate products.

Although the overall research fit in the journal, there are several points that need to be addressed. More importantly, I think that a better rationale for this works is necessary as well as some changes in the structure and/or a reduction in the length. In view of this, I suggest mayor revisions. I do really hope that this review will be useful to the authors to improve their manuscript.

A: We thank the reviewer for the feedback, we will try to address each one of the following points, and to do as much as possible of the suggested modifications.

Q: It is not clear why is important to use these statistical models. Extreme events and its impacts (e.g. February 2018) are barely mentioned in the Introduction and also the limitations in represent snowfall events of the Climate models using emission scenarios. What is the main goal of applying these methods? For the experiment presented, it seems that is to find a better representation in climate change models scenarios. But, It is applicable to weather forecast as GFS for instance? It would be useful to have a better idea of why this work is important. To extend the rationale of the work presented.

A: These models are not meant to be applied to weather forecasting, but rather to correct or complete two types of data: 1) observational dataset where temperature and precipitation are available, but snowfall is not; 2) climate projection data. Even though climate projections are obtained with deterministic models of the atmosphere like weather forecasting, the latter is not characterized by the same level of uncertainty as secular simulations; weather forecasting already incorporates more complex statistical/machine learning algorithms to approximate local weather (for example, as in the output of weather apps). Concerning the extremes: we mention compound extremes as their study has been our first motivation to work with snowfall and to find a way to
reconstruct it in a more accurate way. However, it is not necessarily the case where a compound weather extreme only involves a very extreme daily snowfall, indeed the ensemble of weather conditions determines the severity, including the persistence of several snowy days. For this reason, we looked for a method able to correct the entire probability distribution, thus including but not limited to the improvement of extremes. We observe the capability of the models to improve extremes in the better representation of the snowfall distribution tails in Fig\_Alps\_2 and Fig\_Norway\_2 (see attachments). In the text, this is mentioned at lines 707-8. This is indeed a very synthetic statement, due to the fact that the figure is quite self-explanatory in this sense.

In summary, we briefly mention compound extremes as our own motivation, while in the text we try to make it clear that this method could be used also in case of hydrological studies based on data and analysis of climate projections of snowfall, which is why we do not give a specific focus to extreme snowfall events, but to the variable as a whole.

Q: Some sections are hard to read. The Introduction seems more appropriate for a review paper, maybe avoid the details of each earlier model. Method section also seems lengthy and hard to follow, I suggest to try to go to the point. Results also contain information that is not appropriate for this section (see below). I suggest re-restructuring the manuscript, and move some sections to a Supplementary Material in order to reduce the length of the manuscript.

A: We agree about the length of the paper, we will move the lengthy Subsections 3.1.2-3.1.4 about statistical methodology to an Appendix, with the exception of the first paragraph of 3.1.2, which will be modified to:

In order to overcome the limitations of the STM of  $f_s$ , we aim at reproducing the potentially nonlinear relationship between T and  $f_s$ . As already mentioned in Section \ref{int1}, we decide not to adopt NLS to directly fit S-shaped functions to the data. Other than the sensitivity of this methodology to the optimization algorithm and to initial values, we envisage two more reasons to avoid direct S-shape fitting. First,

**NHESSD**
there is no prior indication of which among the possible S-shaped functions (logistic, hyperbolic tangent, sigmoid) is universally better; this would require to compare the fit and the predictive performance of different specifications at each grid point. Moreover, the points separating the asymptotically horizontal regimes from the centre of the S-shaped curve can be seen as corresponding to two temperature thresholds analogous to  $T_{\text{low}}^{*}$  and  $T_{\text{high}}^{*}$  in \cite{pipes1977ubc}. While the values of these threshold temperatures carry interesting information, it is not immediate to retrieve them from the specified S-shaped functions, nor it is to inform the NLS estimates with these threshold values if their estimate is available.

We propose a way to extend the method by \cite{pipes1977ubc} using a two-step approach: \begin{enumerate} \item[I] determine the optimal number \$m\$ of thresholds temperatures for each grid point and their value, using a breakpoint search algorithm; \item[II] in each of the \$m + 1\$ regimes corresponding to the estimated \$m\$ thresholds, we describe the relationship between \$T\$ and \$f\_s\$ using a regression model. \end{enumerate}

In Appendix \ref{appx}, we describe the search algorithm used to determine the threshold temperatures at each grid point, and we show three different ways to perform MTR: segmented regression on logit-transformed data, beta regression with logit link function, and spline regression on logit-transformed data.

Q: Maybe a Discussion section will do the manuscript more clearer moving some Results and Conclusion paragraphs to this section, allowing concentrate (and reducing) the Conclusions to the main findings.

A: Thank you for the suggestion, we will rename the Conclusions section to Discussion (and we will add a Limitation paragraph as suggested by another reviewer) and we will add a brief Conclusions section:

We have presented two statistical methods equally effective in estimating the snowfall fraction of total precipitation, provided that a reliable measure of near-surface temper-

NHESSD
ature is available. This is a relevant problem in both hydrology and climatology, since an accurate estimation of snowfall is a troubled objective in case of both observed or simulated precipitation.

Both model are an extension of traditional precipitation phase partitioning methods based on estimating the snowfall fraction of total precipitation on the base of one or multiple threshold temperatures. We estimate such thresholds by means of a breakpoint search algorithm.

The two model perform better than their more traditional competitors in terms of prediction error and correlation between real and reconstructed values in a train-test sets validation framework based on the ERA5 reanalysis dataset, showing robustness to climate change. When applied to reconstruct the snowfall in a regional circulation climate model, both techniques produce results with a markedly reduced bias respect to ERA5, when compared to raw climate model simulations.

We conclude that statistical models based on segmented linear or spline regression and informed by bias corrected temperature and precipitation are capable of providing a reliable reconstruction of snowfall that can replace more complex or non-feasible bias correction technique, with better performances than similar models based on parametric assumptions or binary separation.

Q: Additionally, please note that some literature refers to the "separation of snow-fall" as "precipitationâ ÌĘAËĞRphase partitioning methods (PPMs)" (e.g. Harder and Pomeroy,2014). Consider in to use this terminology.

A: We will mention this terminology in the same paragraph interested by your comment about lines 58-59 (see below) so that readers familiar with the term can find the article, if interested. We also use the term in the new brief Conclusions section.

Q: Other comments (line number indicated):

16: This was already mentioned in Line 12.25-30: Add a more specific statement
about why statistical models are better than physics parameterization or simplifications in climate models. Is this just a thought? Or Do you have more evidence of this? some references?

A: We are not sure we correctly understand the line number citation in this question, but concerning the statements at lines 25-30, which seems to be the problem:

We do not mean to state that statistical models are better than climate models due to physical parameterizations; we point out how a statistical step is usually required to perform a bias correction on variables simulated in climate models. Usually this step is a BC as described later in the section, but for snowfall this cannot be conducted effectively without massive complications. So our strategy is to use a well established framework, i.e. the reconstruction of snowfall from T and precipitation, to replace the BC step of snowfall, which is a derived quantity affected by more problems than total precipitation.

To avoid the confusion, we will reformulate the paragraph to:

"In this manuscript we explore the possibility of reconstructing snowfall from bias corrected temperature and precipitation via adequate statistical models, to obtain an improved estimate compared to raw snowfall simulation from the climate model.

Climate Models are the primary tool to simulate multi-decadal climate dynamics and to generate and understand global climate change projections under different future emission scenarios. Both regional and global climate models have coarse resolution and contain several physical and mathematical simplifications that make the simulation of the climate system computationally feasible, but also introduce a certain level of approximation. This results in statistical biases that can be easily observed when comparing the simulated climate to observations or reanalysis datasets."

Q: 50: "Hemispheric".

A: Thank you, we will correct this typo
Q: 58-59: Worth to mention that solid precipitation also depends on relative humidity and could be useful to estimate the mixed-phase or sleet (e.g. Ding et al., 2014).

A: We will add this consideration modifying the text as follows:

Besides the discontinuity of snowfall fields - a feature in common with total precipitation - snowfall is the result of complex processes which involve not only the formation of precipitation, but also the existence and persistence of thermal and hygrometric conditions that allow for the precipitation reaching the ground in the solid state. As a result, snow is often observed in mixed phase with rain, especially when considering daily data. This phase transition poses additional challenges to the bias correction of snowfall, namely the need of separating the snow fraction, using the available meteorological information. Ideally, methods to perform such separation, also known as precipitation phase partitioning methods, should be based on wet-bulb temperature, to which the snow fraction is particularly sensitive in the case of mix precipitation/sleet (Ding et al., 2014). However, due to the difficulty to estimate this parameter in the case of climate models, the task is usually performed relying on temperature data.

Q: 116: Change "Section ??" to "Section 4".

A: Thank you, we will correct this typo

Q: 125-128: This paragraph sounds like a better justification of the work and fits with the overall aim of the journal. Consider to move to the Introduction section and explain a bit more the justification of this study.

A: Thank you, we will move this paragraph to the Introduction section.

Q: 134-135: The election of these months is a bit contradictory if the aim is to analyse extreme events causing disruption. Of course, extreme events largely occur in winter, but over other seasons (end of Autumn, the beginning of Spring) extreme snowfall events could occur. For instance, the so-called "Beast from the East" in 2018 was between the end of February and beginning of March. If analysis months is not extended
maybe change the title to "Improving winter snowfall representation..."

A: We agree that adding more months would be more complete. However, since we will apply this method to an ensemble of climate projections including off-winter months, which will indirectly give also a further validation, we think it is more practical to simply change the title as suggested.

Q: 147: Delete extra "()".

A: Thank you, we will correct this typo

Q: 241: Numbering missing.

A: Thank you, we will correct the numbering of this section, which will also be moved to an Appendix.

Q: 246: Delete extra "()".

A: Thank you, we will correct this typo

Q: 509: Fig. 2 is mentioned here before Fig. 1. Also change to "Fig. 2" to be consistent with the rest of the manuscript.

A: Thank you, we will correct this typo and the order of the figures.

Q: 526: You refer here to Fig. 2 I guess not Fig. 1.

A: Thank you, we will correct this typo

Q: 551: Most of section 4.2 seems more appropriate to the Methods section (or Supplementary Material). For instance, lines 575-585 is an explanation of the Kruskal-WallisH test, lines 593-601 is an explanation of the U-test. Here, you must show the results after applying these tests.

A: Thank you, we will move these explanations to the technical appendix

Q: 536: Numbering missing.
A: Thank you, we will correct this typo

Q: 737-755: These paragraphs are a repetition of previous statements already presented in the Introduction and related to limitations of previous methods, observational data and the physics of the climate models. I think is not necessary to repeat in the Conclusions.

A: These sentences will no longer be in the Conclusions section, which is now much shorter as explained above. If the referee agrees, we would leave them in the now Discussion section, where we think it could be useful to summarize these concepts, given that the paper is quite long.

Q: Figure 7d): Legend is not visible. Figures: please add units and names to the axis where these are missing.

A: Thank you, we will correct these issues with the figures.

**NHESSD**

---

## Author Comment (AC3) · 14 Mar 2021

The manuscript by Pons and Farada assesses the performance of several snowfall separation methods to reproduce simulated snowfall in the ERA5 reanalysis on a European scale by taking into account simulated near-surface air temperature and total precipitation at daily resolution. The two best-performing methods are in a second stage applied to bias-adjusted output of the IPSL-WRF regional climate model (historical period) to obtain a bias-adjusted estimate of simulated snowfall in the RCM. The

evaluation reveals a satisfying representation of the PDF of the daily ERA5 reference snowfall amount in the historical period by the bias-adjusted and separated IPSL-WRF simulation. Overall, the paper fits well into the journal's scope. Data and methods are for most parts clearly introduced and explained. The presentation of the results has some weaknesses but is still acceptable. The major drawback of the work, however, is the unclear relevance of the work for a broader audience and for RCM snowfall bias-adjustment. Essentially, the authors search for a method to emulate the ERA5 micro-physics scheme that simulates the actual snowfall flux in the reanalysis model taking into account simulated near-surface temperature and simulated total precipitation only. The two best performing methods are then applied to a different model (IPSL-WRF) to separate snowfall from total precipitation after bias-adjustment of simulated temperature and precipitation. Results look satisfying, but there is

Q: (1) no evaluation of the ERA5 snowfall flux (which is the basic reference in the entire work, and the entire analysis is geared towards a reproduction of ERA5-simulated snowfall flux; the paper frequently uses the term "observed" for ERA5 snow-fall flux,although it is essentially a simulated flux probably subject to systematic biases)

A: We agree that the word "observation" is incorrectly used to describe ERA5. We corrected all sentences containing such inaccuracy, and we will instead use the term "reanalysis" or the expression "reference dataset".

It is indeed possible that ERA5 presents some biases compared to observations, even though it is not in the scope of our paper to evaluate the accuracy of ERA5 with respect to direct measurement.

In general, the bias correction of climate projection models with respect to observations or reanalysis is a well established practice. Reanalysis datasets such as ERA5, ERA-interim or NCEP are often considered as reference datasets in this context, even if it is known and accepted that they have limitations, as these are generally balanced by the advantages.

Q: (2) no analysis to what extent the satisfying results of the application of the method to the RCM are specific for the chosen RCM and the bias-adjustment method of temperature and precipitation that was carried out beforehand (a different RCM might, even after bias-adjustment, have a completely different multivariate structure of daily temperature and precipitation, at least a structure that is different to ERA5, and the method might not hold in these cases

A: We agree about the fact that our results cannot be generalized to different types of BC used to adjust the RCM. We remark that BC is a computationally expensive and very time consuming operation, and very rarely one can try and compare several different types of BC in a climate study. In our specific case, we considered a model bias adjusted with univariate CDFt, as mentioned at lines 154-7. This method has been widely applied and validated, and it has been used to prepare the datasets constituting the CORDEX-Adjust project, from which we downloaded the already bias adjusted output.

Our method requires the use of an effective BC method for temperature and precipitation beforehand, in the same way it would require well calibrated measuring stations if we were dealing with in-situ observations. Unfortunately, while official measuring stations are regulated by WMO standards, there is not a BC method considered a universal standard. Exploring several BC methods in this study, their multivariate performance and its impact on catching the microphysics over the reference period would be a very heavy task which goes beyond our objectives, and it would produce an extremely large amount of supplementary data.

As a further consideration, we remark that all of the methods representing snowfall with the same philosophy require the knowledge of temperature and precipitation, so the same objection should be true for all the empirical methods already existing in the literature and cited or even put to the test in the present paper.

Overall, we consider it impossible to evaluate several BC methods and their multivariate

impact as a part of our study, as this step alone would constitute a completely different (and probably larger) paper. In a similar way, also testing a model ensemble would be beyond the scopes of this paper, and would still not be exhaustive. For example, if we considered the entire CORDEX ensemble, this would still not make the results directly transferable to CMIP5 models.

We will add a "Limitations" paragraph to the Conclusions section concerning issues with datasets and BC methodology:

"Limitations

We also clarify some of the limitations of our analysis. The nature of climate datasets makes multiple comparisons among models and BC techniques very demanding in terms of data storage and computational time. For this reason, we limited our analysis to one reanalysis dataset (ERA5), one marginal bias correction technique (CDF-t), and one climate projection model (IPSL$\_$WRF).

We do not consider the choice of ERA5 problematic with respect to other gridded datasets that could be observational (e.g. E-OBSv20) or other reanalysis (e.g. NCEP/NCAR): while the actual values could change between datasets, we do not foresee this affecting directly the performance of the methodology we presented in terms of improvement of raw simulations respect the chosen reference dataset.

On the other hand, the choice of the BC may influence the outcome of our modelling procedure. The CDF-t is applied marginally to each variable, so that there is no guarantee that the inter-variable correlations are correctly reproduced in the target climate simulation. Indeed, \cite{meyer2019effects} showed that applying multivariate as opposed to univariate BC produces significant changes in estimated snow accumulation, stressing the importance of modelling the interdependence between precipitation and air temperature in hydrological studies focused on snowy areas. The choice of the BC, in general, should be tuned on the trade-off between complexity and need for controlling specific features, in this case inter-variable correlation. In our case, we con-

sidered a climate dataset prepared in the context of the CORDEX-Adjust project, which is made available already adjusted with respect to ERA5 using marginal CDF-t. Our results show an improvement in snowfall representation even relying on marginal BC; however,we stress that the methodology should be validated again if used on datasets prepared with different BC techniques, to assess whether this difference affects the predicting performance of the model.

On the same note, we remark that prediction accuracy may vary across different climate models, due to the different physical approximations and parameterizations, which are likely to affect the relationship between near-surface temperature and precipitation. Due to these differences, even other RGMs from the EURO-CORDEX project may exhibit variability in the performance of the snowfall reconstruction. This holds true for all statistical models cited in Section\ref{int1}, as it is rarely the case that snowfall reconstruction techniques are tested over an ensemble of different climate models. Once more, we underlying the importance of assessing the performance of the chosen methodology to approximate snow (or compare several of them) by validating it on the historical period of the available models in reference to the available reanalysis/observation dataset."

Q: (3) no discussion of potential problems with inter-variable dependencies even after bias-adjustment of an RCM (-> see, for instance, Meyer et al., HESS, 2019 Effects of univariate and multivariate bias correction on hydrological impact projections in alpine catchments)

A: We thank the reviewer for this suggestion, we included this in the Limitation section mentioned above.

Q: (4) no indication if the identified methods will also produce robust snowfall estimates in a future climate change scenario (which is, as far as I can guess, the basic motivation of the entire work -> a possibility to investigate such an applicability would be to split the ERA5 period into "cold" and "warm" years and to calibrate on the cold and validate

on the warm sample)

A: We thank the reviewer for suggesting such an interesting and yet feasible addition to our validation procedure. We performed the suggested experiment and summarized the results in Fig_warmcold.pdf (see attachments). We will add the following paragraph to the manuscript:

"\Subsection{Robustness to climate change}

As an additional element to evaluate the performance of the identified methods, we assess if they can produce robust snowfall estimates in a climate change scenario. In order to do so, we repeat the validation procedure described in Section \ref{design}; after ordering the ERA5 dataset based on the annual DJF average temperature, we take the coldest $25\%$ as the train set and the warmest $25\%$ as the test set. We run this procedure for the two best performing models, the segmented linear regression and the cubic spline regression.

Fig. \ref{warmcold} shows the model performance metrics in analogy with Figures \ref{boxplots} and \ref{boxcor}. Panel (a) and (b) display the map of the event-to-event correlation coefficient, showing overall higher value than for the random train and test sets. The two models also perform much more similarly in terms of correlation than in the random sets case, as it can be seen from the boxplots in panel (c); the performance in terms of RMSE and MAE is also comparable (panel (d)), as it was in the overall validation presented before. Overall, assuming that separating cold and warm year can be a proxy of climate change to assess model performance, the two technique perform very similarly to the general case in terms of forecasting error, without any visible improvement or accuracy decrease. However, we observe an improvement in the correlation between predicted and true forecasting values: we argue that this effect is likely due to precipitation patterns in years characterized by extreme temperatures in the historical period, and it should not be expected to happen under future climate change."

Q: (5) no reference to differences in spatial resolution of the models employed and the fact the subgrid orography can actually have a considerable influence on simulated snowfall (or, the other way round, neglecting subgrid scale orographic variability in model bias-adjustment could results in false derived snowfall sums)

A: Concerning the difference in spatial resolution of the adopted models, all the dataset we consider have the same lon-lat grid with $0.25°$ resolution, so there are no differences in resolution to be considered in the data we used.

It is plausible that the performance of the algorithm could change if we considered datasets with a resolution sensibly different from the one chosen here, for example 1 km or 100 km. However, comparable methods, including the ones cited here, are applied to anything from a single station time series to gridded datasets without necessarily exploring all possible scales.

The reviewer also underlines that "subgrid orography can actually have a considerable influence on simulated snowfall" and that "neglecting subgrid scale orographic variability in model bias-adjustment could result in false derived snowfall sums". We are aware of the limitation of neglecting subgrid scales, but this is something we always have to live with when dealing with climate simulation models. Considering the lack of scale separation in the atmosphere, the correct description of any phenomenon would benefit from including more fine scales, but sometimes this is not possible. Indeed, Frei et al. 2018 underline that the choice of more complex functional form (which we replace with segmented and spline regression) instead of the binary threshold separation is made precisely to the purpose to approximate subgrid effects.

Q: (6) no analysis of a calibrated threshold within the "naive" STM method (which I assume could yield even better results than the two best identified methods, as even the performance with a fixed 2åŮ̧C threshold is very close to the two best-performing methods)

A: Indeed, we chose not to try a finer calibration of the threshold for the STM model.

This is because, dealing with a high number of grid points, such calibration would still require an explorative technique such as a breakpoint search. However, the STM model with threshold determined in such a way, would correspond to the case of our spline regression, but constraining the number of thresholds at 1 and using 0th-order splines. Since admitting up to two thresholds and cubic splines is not more complicated or sensibly more time demanding, we did not think it is worth to add these constraints: where the optimal model would be a binary threshold, our algorithm can still reproduce that feature while being more versatile if more complex parameterizations are needed.

We do not agree that the STM result is "very close" to the best performing model: it is somehow halfway in terms of average values (both for error measures and correlation) but showing a high variability. This is likely due to the fixed threshold, but as we explained above, it makes no sense from a practical point of view to test this method with more complex thresholds. In fact, such a method is used (for example, in Frei et al. 2018; Bai et al 2019) in the literature taking $2°C$ as an accepted, overall well working typical threshold. As we mentioned in the manuscript at lines 205 and following, a sensitivity analysis over Europe has been conducted, for example, in Faranda 2020, finding that thresholds varying between 0 and $2.5°C$ produce rather comparable results.

Q: (7) no analysis of the importance of variations on the sub-daily scale which might be important for daily snowfall sums.

A: We agree that including small scales in space and time would improve the representation of snow, but we stress again that it is not typical to deal with sub-daily datasets of long term climate projections. Even gridded observations datasets, such as E-OBS, are provided at the daily frequency, making such an evaluation de facto impossible (see e.g. Bai et al. 2019).

Q: The main message of the manuscript is currently, that for this specific setup (this specific RCM, this specific bias-adjustment method, this specific reference snowfall),

the two identified methods if applied to bias-adjusted IPSL-WRF temperature and precip-itation output can yield a representation of snowfall that well reproduces the ERA5reference snowfall. These results are in my opinion not per se transferable to different models or to a future climate scenario or to a different reference snowfall (especially not to a true observation-based snowfall estimate). As such, the value of the work is limited for the time being in my opinion and not too informative for a broader readership.I would hence recommend to return the manuscript to the authors for major revisions. During these revisions, the mentioned points should be picked up in order to increase the relevance of the work. A couple of further issues are mentioned below. With kind regards.

A: As already mentioned in response to Reviewer's point 2, these issues are now explicitly mentioned in the Limitations paragraph in the Discussion section. We stress that we agree that all these limitations exist, but we find that it would be hardly feasible to test one statistical method by varying: Time resolution Space resolution Reference dataset Bias correction technique Physics and numerical climate model schemes

To our knowledge, such a broad validation does not exist even for simple and well established models (e.g. single threshold binary separation). As far as we can tell, most of the literature dealing with this type of statistical model for precipitation phase apportionment are validated on datasets similar to the ones we considered in this paper. We think that the Limitations section should make it clear that the results shown in the paper should be considered specific to our setting, and that a validation of the model should be performed, if this is applied to different datasets.

FURTHER ISSUES:

Q: Line 24: Very unclear what is meant.

A: We agree that this sentence was not clear. We replaced it with a simple example of a case where the binary apportionment could produce a severe bias:

"Even though such binary separation of snowfall using a temperature threshold seemed a good option to retrieve snowfall data from E-OBSv20.0e, it has obvious limitations: for example, in an event characterized by abundant precipitation but a temperature associated to a roughly 50$\%$ snow fraction, snowfall would be either strongly under- or overestimated."

Q: Lines 35-36. Also rather unclear.

A: We agree about lack of clarity and we also realized this sentence was quite redundant. We will remove lines 35-38 and change the next sentence to better match the previous paragraph to:

"In order to mitigate the aforementioned biases, a BC step is usually performed. This step usually consists of a methodology designed to adjust specific statistical properties of the simulated climate variables towards a validated reference dataset in the historical period. [...]"

Q: Lines 38-40: This is actually not true, the entire set of so-called "perfect prognosis" downscaling methods is ignored here. These do not adjust the simulated variables towards observations but exploit calibrated relationships between observed (or reanalysis-simulated) large scales and observed local scales.

A: In agreement with the existing literature, we consider perfect prognosis downscaling as a part of statistical downscaling (see, e.g. Soares et al., 2019)

Q: Lines 141-142: Very unclear.

A: We will change this sentence to:

"This quantity is relevant for hydrologists, being closely related to runoff and river discharge, but also for climatologists, since it well represents the intensity of the phenomenon while, however, we remind that snowfall is not a measure of accumulation of snow on the ground."

Q: Line 145: Above (line 132) you mention that only daily data are used, here you obviously employ hourly data. Please clarify.

A: We better clarify changing the sentence at line 145 to:

The initial ERA5 dataset is available at hourly frequency, while the IPSL$\_$WRF is available at daily frequency. Since the two time steps are different, and we have no way to disentangle a daily quantity into the sub-daily cycle, we aggregate the hourly ERA5 data into daily.

Q: Line 151: "grid step" unclear

A: We will change the sentence to: In particular, we consider DJF data from climate simulations of the historical period 1979-2005 over the same domain and at the same spatial resolution as the reanalysis dataset described in Section \ref{era5mod}.

Q: Line 181: Rather unclear what is meant by "standardized temperature anomalies" and why these are used.

A: We now specify the standardization procedure as follows: "In all the regression models discussed in the following, but not in the STM, we use as independent variable the standardized temperature anomalies, obtained by subtracting the historical mean and dividing by the historical standard deviation."

Using standardized anomalies is quite a common practice in climate studies, where different variables span over very different scales (e.g. precipitation is in average about 0.1 m/day, absolute temperature is of order 250-300 K, geopotential height 5000 m). For transparency, we report to the reviewer that, in particular, we had standardized the variables as we tried to add total precipitation as a covariate, to consider the possible influence of intense precipitation on the snow fraction. Given the very different scale between the two covariates (temperature and precipitation) we standardized the variables in the various model specifications. Improvement obtained by adding precipitation were so unremarkable (practically non existent) that we did not even mention

them in the manuscript, and kept the results of the models based on temperature with standardized variables.

We do not foresee affecting our results and it is quite a common practice in regression modelling even outside climatology. As we already stressed at line 182, it is important to do the standardization in the same way to reproduce the result.

Q: Chapter 3.1.2: This sub-chapter contains a large amount of rather technical information, which is appreciated, but which should be moved to some technical appendix I believe.

A: We agree and we will move this subsection to a technical appendix.

Q: Line 517: Do you have any explanation for these rather low calibrated thresholds? Is there a relation to orographic height, for instance?

A: We do not have an explanation for this. We noticed that such low thresholds appear in areas where we would expect winter precipitation to be mainly snow regardless of the specific daily temperature, given the cold continental or subarctic climate. The search algorithm is not meant to necessarily find physically meaningful values, so it is possible that it finds thresholds that improve the performance only very slightly, while even a regression without any threshold or even a binary apportionment could perform relatively well. In this sense, the threshold recovered after the recomputation via the segmented regression algorithm make more sense (temperature over continental areas seem positive, but here please consider that we are looking at anomalies).

Q: Line 526: Should be "Fig. 2" instead of "Fig. 1".

A: Thank you, we will correct accordingly

Q: Lines 688-689: Very unclear.

A: We will change this sentence to:

Since comparing IPSL$\_$WRF and its adjusted versions to ERA5 does not provide

a one-on-one correspondence between snowfall events, it is not possible to compute correlation coefficients between reanalysis and model snowfall time series at each grid point as in Fig. \ref{boxcor}. Instead, we can study the correlation between the total 1979-2005 ERA5 snowfall and the total 1979-2005 snowfall simulated by IPSL$\_$WRF and approximated with segmented logit and cubic spline regression at each grid point. Fig. \ref{alps\_dist} (a) shows the scatterplots of total IPSL$\_$WRF, logit segmented regression and cubic spline regression snowfall against total ERA5 snowfall for the grid points in the Alps region.

Q: Lines 707-708: Better representation of the tails is not really apparent from the figure I'd say.

A: Even though we are aware that the definition of "tail" is somehow arbitrary, we remark that when the ERA5 distribution hits the 0.95 mark, the IPSL-WRF distribution function is barely above 0.85, and when ERA5 reaches 0.99 IPSL-WRF is around the 0.95. On the other hand, the two statistical models are practically non distinguishable from ERA5, considering that this holds true for values above the 95th percentile, we think that the improvement in the tail is rather solid.

Q: Figure 1: Color scale is not very intuitive.

A: We changed the palettes to more traditional ones. The initial choice of a palette alternating different color was due to the fact that the total snowfall spans several orders of magnitude between the most and least snowy locations on the map, so that less contrasting color scales tend to flatten the variability.

Q: Figures 2 and 3: Bad color scale: White color means threshold temperatures around 0â̊ǫC but also "not applicable". I'd suggest to modify the color scale.

A: We agree. We will change the color associated to 0°C to gray, to make the figures clearer.

Q: Figure 4: Legend too large. Also, the methods are named differently compared to

Table1 and are sorted in a different order. Please harmonize. Also, it would be good to use the same unit in the lower panel as in Table 1 (10Ë$\mathrm{E}$-3)

A: We proceeded to make these modifications to the figure.

Q: Figure 5: Upper panel: Please use the same sorting of methods as in Table 1.

A: We proceeded to make this modification to the figure.

Q: Figure 6: Very bad color scale, not at all intuitive. Also, the color scale should be identical for all panels to enable a comparison (same color should mean the same value in all panels). Is the unit actually m/27 years (1979-2005) or m/year? Please clarify.

A: We proceeded to change the palette with one more traditionally used for anomalies. Unfortunately, setting the colorscale in such a way that the same color has the same value over the plot would completely flatten the aspect of panels b) and c), since values in panel a) can be one order of magnitude larger.

Q: Figure 7: Legend of lower panel too small.

A: We proceeded to make this modification to the figure.

Q: Figure 8: What about the bad-performing grid cell in northern Italy in logit seq and cubic spline? What is happening here?

A: We inspected the specific grid cell in detail to assess the extreme negative value. Indeed, it seems to be one of the few points where the breakpoint search algorithm failed to converge, so that a threshold is not available and the models were then not estimated. When we took sums over time to compute snowfall totals, an 'na.rm = TRUE' option was used in the R script so that instead of an NA the sum in that grid cell resulted equal to 0, and the difference was then the negative ERA5 snowfall total in that cell. We ran the scripts without the NA removal option and produced a new figure with the concerned grid point correctly masked out as NA.

Q: Figures 9 and 11, upper panels: Sorry, but even after reading the explanation several times it is not really clear to me what is displayed here. Also, I'd suggest to use a white background instead of a black background. Lower panels: Please specify the unit of the x-axis

A: We proceeded to make this modification to the figure.

References

Soares, P. M., Maraun, D., Brands, S., Jury, M. W., Gutiérrez, J. M., San‐Martín, D., ... & Obermann‐Hellhund, A. (2019). Process‐based evaluation of the VALUE perfect predictor experiment of statistical downscaling methods. International Journal of Climatology, 39(9), 3868-3893.

Bai, L., Shi, C., Shi, Q., Li, L., Wu, J., Yang, Y., ... & Meng, J. (2019). Change in the spatiotemporal pattern of snowfall during the cold season under climate change in a snow‐dominated region of China. International Journal of Climatology, 39(15), 5702-5719.

———————————————————

[Figure]

**Fig. 1.**

[Figure]

**Fig. 2.**

[Figure]

**Fig. 3.**

[Figure]

**Fig. 4.**

[Figure]

[Figure]

[Figure]

**Fig. 5.**

[Figure]

**Fig. 6.**

[Figure]

**Fig. 7.**

[Figure]

**Fig. 8.**

[Figure]

[Figure]

[Figure]

**Fig. 9.**

[Figure]

**Fig. 10.**

[Figure]

[Figure]

[Figure]

**Fig. 11.**

[Figure]

**Fig. 12.**